# Non-Euclidean Universal Approximation

**Anastasis Kratsios**[*]

**Eugene Bilokopytov**[†]

## Abstract

Modifications to a neural network's input and output layers are often required to accommodate the specificities of most practical learning tasks. However, the impact of such changes on architecture's approximation capabilities is largely not understood. We present general conditions describing feature and readout maps that preserve an architecture's ability to approximate any continuous functions uniformly on compacts. As an application, we show that if an architecture is capable of universal approximation, then modifying its final layer to produce binary values creates a new architecture capable of deterministically approximating any classifier. In particular, we obtain guarantees for deep CNNs and deep feed-forward networks. Our results also have consequences within the scope of geometric deep learning. Specifically, when the input and output spaces are Cartan-Hadamard manifolds, we obtain geometrically meaningful feature and readout maps satisfying our criteria. Consequently, commonly used non-Euclidean regression models between spaces of symmetric positive definite matrices are extended to universal DNNs. The same result allows us to show that the hyperbolic feed-forward networks, used for hierarchical learning, are universal. Our result is also used to show that the common practice of randomizing all but the last two layers of a DNN produces a universal family of functions with probability one. We also provide conditions on a DNN's first (resp. last) few layer's connections and activation function which guarantee that these layer's can have a width equal to the input (resp. output) space's dimension while not negatively effecting the architecture's approximation capabilities.

## 1  Introduction

Modifications made to a neural network's input and output maps to extract features from a data-set or to better suit a learning task is prevalent throughout learning theory. Typically, such changes are made by pre-(resp. post-)composing an architecture with a fixed and untrainable feature (resp. readout) map. Examples prevail classification by neural networks, random feature maps obtained by randomizing all but the last few layers of a feed-forward network, and numerous illustrations throughout geometric deep-learning theory, which we detail below. This motivates the central question of this paper: *"Which modifications to the input and output layers of a neural network architecture preserve its universal approximation capabilities?"*

Specifically, in this paper we obtain a simple sufficient condition on a pair of a feature map $\phi : \mathscr{X} \to \mathbb{R}^m$ and a readout map $\rho : \mathbb{R}^n \to \mathscr{Y}$, where $\mathscr{X}$ and $\mathscr{Y}$ are topological spaces, guaranteeing that if $\mathscr{F}$ is dense in $C(\mathbb{R}^m, \mathbb{R}^n)$ for the uniform convergence on compacts (ucc) topology then

$$\{f \in C(\mathscr{X}, \mathscr{Y}) : \rho \circ f \circ \phi, f \in \mathscr{F}\}, \tag{1}$$

---

[*]Department of Mathematics, Eidgenössische Technische Hochschule Zürich, HG G 32.3, Rämistrasse 101, 8092 ürich, Switzerland. email: *anastasis.kratsios@math.ethz.ch*

[†]Department of Mathematics and Statistical Sciences, University of Alberta, 11324 89 Ave NW, Edmonton, AB T6G 2J5, Canada. email: *bilokopy@ualberta.ca*

is dense in $C(\mathscr{X},\mathscr{Y})$ in the uniform convergence on compacts topology when $\mathscr{Y}$ is metric and, more generally, in the compact-open topology when $\mathscr{Y}$ is non-metrizable (such as in the hard classification problem). Simplified conditions are obtained when $\mathscr{Y}$ is a metrizable manifold, and characterization of $\rho$ and $\phi$ is obtained when both $\mathscr{X}$ and $\mathscr{Y}$ are smooth manifolds.

The set $\mathscr{F}$ represents any expressive neural network architecture. For example, by [35] $\mathscr{F}$ can be taken to be the set of feed-forward networks with one hidden layer and continuous, locally-bounded, and non-polynomial activation function. Or, by [56], $\mathscr{F}$ can be taken to be the set of deep convolution networks with specific sparsity structures and ReLu activation function. Throughout $\mathscr{F}$ is often referred to as an architecture. The results are not limited to neural networks and remain valid when, for example, $\mathscr{F}$ is taken to be the set of posterior means generated by a Gaussian processes universal kernel, as in [41]. The central results are motivated by the following consequences.

### Implication: Method for Constructing Non-Euclidean Universal Approximators

A natural hub for our results is in *geometric deep learning*, an emerging field of machine learning, which acknowledges and makes use of the latent non-Euclidean structures present in many types of data. Applications of geometric deep learning are prevalent throughout neuroimaging [16], computer-vision [48], covariance learning [40], and learning from hierarchical structures such as complex social networks [34], undirected graphs [43], and trees [50].

For instance, in [44], it is shown that low-dimensional representations for complex hierarchical structures into hyperbolic space outperform the current state-of-the-art high-dimensional Euclidean embedding methods due to the tree-like geometry of the former. Using the theory of gyro-vector space, introduced in [55], [17] proposed a hyperbolic-space variant of the feed-forward architecture and demonstrated its superior performance in learning hierarchical structure from these hyperbolic representations. A direct application of our main result confirms that this non-Euclidean architecture can indeed approximate any continuous function between hyperbolic spaces.

More generally, we obtain an explicit construction of feed-forward networks between any Cartan-Hadamard manifold and a guarantee that our construction is universal. Cartan-Hadamard manifolds appear throughout applied mathematics from the symmetric positive-definite matrix-valued regression problems of [16, 40], which we extend to universal approximators, to applications in mathematical finance in [23], Gaussian processes in [39], information geometric in [4], and to the geometry of the Wasserstein space [36] commonly used in Generative Adversarial Networks as in [3].

### Implication: Universal Approximation Implies Universal Classification

Perhaps the most commonly used readout maps are those used when modifying neural-networks to perform classification tasks. The currently available theoretical results, found in [15], guarantee that for a random vector in $\mathbb{R}^m$ with random labels, the set of feed-forward networks with one hidden layer, step activation function $\sigma(x) = I_{[0,\infty)} - I_{(-\infty,0]}$, and readout map $\rho(x)_i = I_{[\frac{1}{2},\infty)}$ can approximate the Bayes' classifier in probability.

As an application of this paper's main results, we obtain deterministic guarantees of generic hard ($n$-ary) and soft (fuzzy) classification on $\mathbb{R}^m$ for any given universal approximator in $C(\mathbb{R}^m,\mathbb{R}^n)$ once it's outputs are modified by a continuous surjection $\rho$ to take values in $\{0,1\}^n$ or $(0,1)^n$, respectively. For example, our result applies to feed-forward networks with at-least one hidden layer holds when $\rho$ the component-wise logistic $\rho(x)_i = I_{[\frac{1}{2},1]} \circ \frac{e^{x_i}}{1+e^{x_i}}$ readout map.

### Implication: DNNs with Randomly Generated First Layers are Universal

We show that the commonly employed practice of only training the final layers of a deep feed-forward network and randomizing the rest preserves its universal approximation capabilities with probability 1. Though widely used, this practice has only recently begun to be studied in [19, 37]. The link with our results arises from an observation made in [37], stating that the first portion of such a random architecture can be seen as a randomly generated feature map.

### Implication: DNNs Can be Narrowed

In [29, 45] the authors provide lower bounds on a DNN layer's width, under which it is no longer a universal approximator. However, there is a wealth of literature which shows that arranging a network's neurons to create depth rather than width yields empirically superior performance. As a final application of our theory, we provide explicit conditions on a DNN's connections and activation functions so additional initial and final few layers may be added to a DNN which do not respect the minimum width requirements of [29, 45] but do not negatively impact the DNN's approximation

capabilities. Numerical implementations show that additional depth build using our main results improve predictive performance additional deep layers failing our assumptions reduced the network's predictive performance.

This paper is organized as follows. Section 2 discusses the necessary topological and geometric background needed to formulate the paper's central results. Section 3 contains the paper's main results discussed above. The conclusion follows in section 4. The proofs of the main results are contained within this paper's supplementary material.

## 2  Background

### 2.1  General Topology

Before moving on to the main results of the paper, we will require some additional topological terminology. The *interior* of a subset $A \subseteq \mathscr{X}$ of a topological space is the largest open subset contained in $A$. For example, in the Euclidean space $\mathbb{R}$, the interior of $[0,1)$ is $(0,1)$. The *closure* of $A$ is the smallest closed-set containing $A$. Therefore, the closure of $[0,1)$ in $\mathbb{R}$ is $[0,1]$. The difference between the closure of $A$ and its interior is called the *boundary* of $A$ and is denoted by $\partial A$. For example, the boundary of $[0,1)$ in the Euclidean space $\mathbb{R}$ is $\{0,1\}$. A subset $A \subseteq \mathbb{R}^m$ is called a *retract* of $\mathbb{R}^m$ if there is a continuous map $r : \mathbb{R}^m \to A$ such that $r \circ \iota_A = 1_A$, where $\iota_A : A \to \mathbb{R}^m$ is the inclusion map and $1_A$ is the identity map on $A$. Dually, a continuous right-inverse of a continuous surjective map is called a *section*. A *covering projection* $f : \mathbb{R}^n \to \mathscr{Y}$ is a surjective continuous function such that for every $x \in \mathscr{Y}$ there is an open set $U_x$, containing $x$, such that $\rho^{-1}[U] = \bigcup_{\alpha \in A} U_x^\alpha$ and $\{U_x^\alpha\}_{a \in A}$ is a disjoint collection of open sets for which $\rho|_{U_x^\alpha} : U_x^\alpha \to U_x$ is a homeomorphism. For example, the map $x \to e^{-i\pi x}$ is a covering projection of $\mathbb{R}$ onto the circle.

Lastly, since continuous maps transfer topological information then a continuous bijection with continuous inverse preserves all topological information. Such a map is called a *homeomorphism* and two topological spaces related by a homeomorphism are said to be *homeomorphic*. For example, the sigmoid (or logistic) function $x \mapsto \frac{e^x}{1+e^x}$ continuously puts $\mathbb{R}$ in bijection with $(0,1)$ and its inverse is the logit function $y \mapsto \ln\left(\frac{y}{1-y}\right)$, which is continuous. Therefore, $\mathbb{R}$ and $(0,1)$ are homeomorphic. A continuous injective map $\phi : \mathscr{X} \to \mathbb{R}^m$ such that $\mathscr{X}$ is homeomorphic to $\phi(\mathscr{X})$ is called an *embedding*.

A topological space $\mathscr{X}$ is said to be *locally-compact* if every $x \in \mathscr{X}$ is contained in an open subset $U_x$ of $\mathscr{X}$ which is in turn contained in an compact subset $K_U$ of $\mathscr{X}$. For example, every point $x \in \mathbb{R}$ is contained in $(x-1, x+1)$ which is in turn contained in the closed-bounded (and therefore compact-set by the Heine-Borel theorem) $[x-1, x+1]$. A topological space is said to be *simply connected*, if every two paths between points can be continuously deformed into one another.

Similarly to [8, 11], we say that a subset $A$ of a topological space $\mathscr{X}$ is *collared* if there exists an open subset $U \subseteq \mathscr{X}$ containing $A$ and a homeomorphism $\phi : U \to A \times [0,1)$ mapping $A$ to $A \times [0,1)$. In this way the open set $U$ is, in a sense, topologically similar to $A$ itself since one can imagine shrinking any point $(a,t) \in A \times [0,1)$ down to $(a,0)$ and then identifying it back with $a$ via $\psi$.

We denote the set of positive integers by $\mathbb{N}^+$.

### 2.2  Topology of Function Spaces

Let $C(\mathbb{R}^m, \mathbb{R}^n)$ denote the set of all continuous functions from the Euclidean space $\mathbb{R}^m$ to the Euclidean space $\mathbb{R}^n$. Closeness in $C(\mathbb{R}^m, \mathbb{R}^n)$ can be described in a number of ways but in the context of universal approximation theorems of [13, 27, 35] two functions $f$ and $g$ are thought of as being close if they are uniformly close on compacts, if for a given $\varepsilon > 0$ the following holds:

$$\sum_{k=1}^{\infty} \frac{\sup_{\|x\| \leq k} \sqrt{\sum_{i=1}^{n} \|f(x)_i - g(x)_i\|^2}}{2^k \left(1 + \sup_{\|x\| \leq k} \sqrt{\sum_{i=1}^{n} \|f(x)_i - g(x)_i\|^2}\right)} < \varepsilon. \tag{2}$$

The topology described by (2) is called the *topology of uniform convergence on compacts*, henceforth ucc topology. If $\mathbb{R}^n$ is replaced by any other topological space $\mathscr{Y}$ whose notion of closeness is defined by a distance function and $\mathbb{R}^m$ is replaced by nearly any other topological space then closeness in the collection of continuous functions from $\mathscr{X}$ to $\mathscr{Y}$, denoted by $C(\mathscr{X}, \mathscr{Y})$, can be described analogously to (2) by replacing the Euclidean distance on $\mathbb{R}^n$ by another distance function $d$-on $\mathscr{Y}$, the compact subsets $K \subseteq \mathbb{R}^m$ with compact subsets of $\mathscr{X}$, and taking $f, g \in C(\mathscr{X}, \mathscr{Y})$. The topology on $C(\mathscr{X}, \mathscr{Y})$ defined in this way is still called the ucc topology.

If a distance function cannot describe the topology on $\mathscr{Y}$, for example, we will see that this is the case for reasonable topologies on $C(\mathscr{X},\{0,1\}^n)$, then one cannot define the ucc topology. Instead, consider the smallest topology on $C(\mathscr{X},\mathscr{Y})$ containing the sets

$$\{V_{K,O} : \emptyset \neq K \subseteq \mathscr{X} \text{ compact and } \emptyset \neq O \subseteq \mathscr{Y} \text{ open}\}, V_{K,O} \triangleq \{f \in C(\mathscr{X},\mathscr{Y}) : f(K) \subseteq O\}. \qquad (3)$$

When the topology on $\mathscr{Y}$ is defined by a distance function and $\mathscr{X}$ is a locally-compact Hausdorff space, then the smallest topology containing (3) coincides with the ucc topology. However, unlike the ucc topology, the smallest topology containing (3) is well-defined on $C(\mathscr{X},\mathscr{Y})$ for any topological spaces $\mathscr{X}$ and $\mathscr{Y}$. This generalized ucc topology is called the *compact-open topology* (co-topology).

### 2.3 Manifolds

A (topological) *manifold* is a topological space which "closeup" resembles Euclidean space, whereas a manifold with boundary locally looks like a part of Euclidean space but possibly with a hard edge.

**Definition 2.1** (Metrizable Manifold with Boundary; [8])**.** *A topological space $\mathscr{Y}$ is said to be a metrizable manifold with boundary if*
  *(i) For every $y \in \mathscr{Y}$, there is an open $U_y \subseteq \mathscr{Y}$ containing y which is homeomorphic to*

$$\left\{ (z_1,\ldots,z_n) \in \mathbb{R}^n : \sqrt{\sum_{i=1}^{n} z_i^2} < 1 \text{ and } z_n \geq 0 \right\}, \qquad (4)$$

  *(ii) There exists a distance function (metric) $d : \mathscr{Y}^2 \to \mathscr{Y}$ such that the topology on $\mathscr{Y}$ coincides with the smallest topology on $\mathscr{Y}$ containing the open balls $\{B_\varepsilon(y)\}_{\varepsilon>0, y\in\mathscr{Y}}$; where*

$$B_\varepsilon(y) \triangleq \{z \in \mathscr{Y} : d(z,y) < \varepsilon\}.$$

*We say that d is a metric for $\mathscr{Y}$. The subset of $\mathscr{Y}$ consisting of all points y contained in some open set $U_y$ which is homeomorphic to the interior of (4) is denoted by $\mathrm{Int}(\mathscr{Y})$.*

A *smooth manifold* without boundary, is a manifold for which there is a well-defined differential calculus admitting arbitrarily many derivatives and which can locally be deformed into Euclidean space via infinitely differentiable maps with infinitely differentiable inverses.

An *m*-dimensional *Riemannian manifold* $\mathscr{M}$ is a manifold without boundary which can be locally smoothly deformed into Euclidean space such that curvature and length can be meaningfully compared, locally, between $\mathbb{R}^m$ and $\mathscr{M}$. Amongst other things, this allows the definition of minimal-length curves connecting points on $\mathscr{M}$, called *geodesics*. If any two points on $\mathscr{M}$ can be connected by such a minimal length curve then $\mathscr{M}$ is said to be *complete*. Moreover, when $\mathscr{M}$ is complete and connected, the function mapping any two points $p,q \in \mathscr{M}$ to the length of a geodesic connecting them defines a metric $d_{\mathscr{M}}$. Thus, $\mathscr{M}$ has a geometrically meaningful metric structure where distance represents the length of maximally efficient trajectories and $C(X,\mathscr{M})$. The existence of $d_{\mathscr{M}}$ also implies that $C(\mathscr{X},\mathscr{M})$ is equipped with the ucc-topology.

Further, when $\mathscr{M}$ is complete and connected the Hopf-Rinow Theorem, of [26], affirms that for any given $p \in \mathscr{M}$, the map sending any $v \in \mathbb{R}^m$ lying tangent to $p$ to the point on $\mathscr{M}$ arrived at time $t = 1$ by traveling along a the geodesic with initial velocity $v$ defines a surjection from $\mathbb{R}^m$ onto $\mathscr{M}$. This map is called the *Riemannian Exponential map* on $\mathscr{M}$ at $p$ and is denoted by $\mathrm{Exp}_p^{\mathscr{M}}$. In [28], it is shown that, in this case, $\mathrm{Exp}_p^{\mathscr{M}}$ has a smooth inverse outside a low-dimensional subset $C_p$. This inverse is denoted by $\mathrm{Log}_p^{\mathscr{M}}$ and is known to locally preserve length between $\mathbb{R}^m$ and $\mathscr{M}$ along geodesics emanating from $p$. This means that $\mathrm{Log}_p^{\mathscr{M}}$ and $\mathrm{Exp}_p^{\mathscr{M}}$ are geometrically meaningful feature and readout maps, respectively.

However, the set $\partial C_p$ can be pathological or difficult to deal with. This issue is overcome by turning to the sub-class of *Cartan-Hadamard manifolds*. A Riemannian manifold $\mathscr{M}$ is Cartan-Hadamard if it is simply connected and has *non-positive curvature*. Non-positive curvature mean that all triangles drawn on $\mathscr{M}$ by geodesics have internal angles adding-up at-most $180°$.

## 3 Main Results

Let $\phi : \mathscr{X} \to \mathbb{R}^m$ and $\rho : \mathbb{R}^n \to \mathscr{Y}$. Subsets of $\mathbb{R}^n$ (resp $\mathbb{R}^m$) will be equipped with the (relative) Euclidean topology unless explicitly stated otherwise. Equip $C(\mathscr{X},\mathscr{Y})$ with the co-topology, $C(\mathbb{R}^m,\mathbb{R}^n)$ with the ucc topology, let $\mathscr{F}$ be a *dense* subset of $C(\mathbb{R}^m,\mathbb{R}^n)$ such as the architectures studied in

[35, 38, 56] or the posterior means of a Gaussian process with universal kernel as in [41], and define the subset $\mathscr{F}_{\rho,\phi} \subseteq C(\mathscr{X}, \mathscr{Y})$ by

$$\mathscr{F}_{\rho,\phi} \triangleq \{g \in C(\mathscr{X}, \mathscr{Y}) : g = \rho \circ f \circ \phi \text{ where } f \in \mathscr{F}\}. \tag{5}$$

The set $\mathscr{F}_{\rho,\phi}$ is dense in $C(\mathscr{X}, \mathscr{Y})$ under the following assumptions on $\phi$ and $\rho$.

**Assumption 3.1** (Feature Map Regularity). *The map $\phi$ is a continuous and injective.*

**Assumption 3.2** (Readout Map Regularity). *Suppose that the readout map $\rho$ satisfies the following:*
 *(i) Either of the following hold:*
   *(a) $\rho$ is a continuous and it has a section on $\mathrm{Im}(\rho)$,*
   *(b) $\rho$ is a covering projection of $\mathbb{R}^m$ onto $\mathrm{Im}(\rho)$ and $\mathscr{X}$ is connected and simply connected,*
 *(ii) $\mathrm{Im}(\rho)$ is dense in $\mathscr{Y}$,*
 *(iii) $\partial\mathrm{Im}(\rho)$ is collared.*

**Theorem 3.3** (General Version). *Suppose that $\mathscr{F}$ is dense in $C(\mathbb{R}^m, \mathbb{R}^n)$. If Assumptions 3.1 and 3.2 hold then $\mathscr{F}_{\rho,\phi}$ is dense in $C(\mathscr{X}, \mathscr{Y})$.*

Just as in the filtering literature of [10], one would hope that the outputs of any learning model should depend continuously on its inputs. Therefore, we only consider feature maps $\phi$ which are continuous functions. In this case, Assumption 3.1 is *sharp*. We denote the identity map $x \mapsto x$ on $\mathbb{R}^n$ by $1_{\mathbb{R}^n}$.

**Theorem 3.4** (Assumption 3.1 is Sharp). *Let $\mathscr{X}$ be a metrizable manifold with boundary, let $\phi$ be continuous, and $\mathscr{F} \subseteq C(\mathbb{R}^m, \mathbb{R}^n)$. Then $\mathscr{F}_{1_{\mathbb{R}^n},\phi}$ is dense in $C(\mathscr{X}, \mathbb{R}^n)$ if and only if $\phi$ is injective.*

**Remark 3.5** (Sharpening Assumption 3.2). *Assumption 3.2 is almost sharp and a characterization can be obtained using the $\mathscr{X}$-sets studied in [54, 20]. However, it is unlikely that a non-pathological example can be generated which falls outside the scope of Assumption 3.2.*

Theorem 3.4 shows that it is easy to verify if a feature map preserves the universal approximation property. However, it can be much more challenging to verify if and when the readout map $\rho$ does so.

The following presents a readily applicable case of Theorem 3.3. They highlight the convenient fact that if $\rho$ is surjective then only Assumptions 3.1 and 3.2 (i) need to be verified.

**Corollary 3.6.** *If $\phi$ is a continuous injective map, $\rho$ is a surjective covering projection, and $\mathscr{F}$ is dense in $C(\mathbb{R}^d, \mathbb{R}^D)$ then $\mathscr{F}_{\rho,\phi}$ is dense in $C(\mathscr{X}, \mathscr{Y})$. In particular, $\phi$ and $\rho$ may be homeomorphisms.*

When both $\phi$ and $\rho$ fully preserve topological structure then Corollary 3.6 sharpens.

**Proposition 3.7** (Homeomorphic case is Sharp). *Let $\phi$ and $\rho$ be homeomorphisms. Then $\mathscr{F}$ is dense in $C(\mathbb{R}^m, \mathbb{R}^n)$ if and only if $\mathscr{F}_{\rho,\phi}$ is dense in $C(\mathscr{X}, \mathscr{Y})$.*

**Corollary 3.8.** *If $\phi$ is a continuous injective map, $\rho$ is a continuous surjection with a section, $\mathscr{X}$ is connected and simply connected, and $\mathscr{F}$ is dense in $C(\mathbb{R}^d, \mathbb{R}^D)$ then $\mathscr{F}_{\rho,\phi}$ is dense in $C(\mathscr{X}, \mathscr{Y})$.*

When additional structure is assumed of $\mathscr{Y}$, as is common in most applications, Assumption 3.2 (ii) and (iii) can be omitted and the other assumptions can be simplified. Specifically, the case where $\mathscr{Y}$ is a manifold with boundary is considered. In the case where $\mathscr{X}$ and $\mathrm{Int}(\mathscr{Y})$ are smooth, then Theorem 3.3 can be further streamlined as follows.

**Assumption 3.9** (Readout Map Regularity: Geometric Version). *Suppose that $\rho$ satisfies:*
 *(i) $\rho$ satisfies Assumption 3.2 (i) and $\mathrm{Im}(\rho) \subseteq \mathrm{Int}(\mathscr{Y})$,*
 *(ii) $\mathrm{Int}(\mathscr{Y}) - \mathrm{Im}(\rho)$ is a (possibly empty) smooth submanifold of $\mathrm{Int}(\mathscr{Y})$ of dimension strictly less-than $\dim(\mathrm{Int}(\mathscr{Y})) - n$.*

**Theorem 3.10** (Geometric Version). *Let $\mathscr{Y}$ be a metrizable manifold with boundary, for which $\mathrm{Int}(\mathscr{Y})$ is a smooth manifold, $\mathscr{X}$ is locally-compact, and $\mathscr{F}$ is dense in $C(\mathbb{R}^m, \mathbb{R}^n)$. If $\phi$ satisfies Assumption 3.1 and $\rho$ satisfies Assumption 3.9 then $\mathscr{F}_{\rho,\phi}$ is dense in $C(\mathscr{X}, \mathscr{Y})$.*

Consequences of these results in various areas of machine-learning are now considered.

## 3.1 Dense Families in $C(\mathbb{R}^m, \mathbb{R}^n)$ Induce Universal Classifiers

Let $\mathscr{X}$ be a set, $\phi : \mathscr{X} \to \mathbb{R}^m$ be a bijection, and $\{L_i\}_{i=1}^n$ be a collection of labels describing elements of $\mathscr{X}$. Let $\mathscr{X}_i \triangleq \{x \in \mathscr{X} : x \text{ has label } L_i\}$. For example, $\{\mathscr{X}_i\}_{i=1}^n$ are disjoint and cover $\mathscr{X}$ then we obtain the $n$-ary classification problem, but in general, any $x \in \mathscr{X}$ may simultaneously have distinct multiple labels. Without loss of generality, we may assume that $\mathscr{X}$ is a topological space which is homeomorphic to $\mathbb{R}^m$ since we may equip it with the topology $\{\phi^{-1}[U] : U \text{ open in } \mathbb{R}^m\}$. Assume that the sets $\{\mathscr{X}_i\}_{i=1}^n$ are open subsets of $\mathscr{X}$.

In the stochastic case, the Bayes classifier is the golden standard for classification. In the deterministic case, the standard is clearly the *ideal classifier* $\hat{h} : \mathscr{X} \to \{0,1\}^n$, introduced here, and defined by

$$\hat{h}(x)_i \triangleq I_{\mathscr{X}_i}(x), \tag{6}$$

where $I_{\mathscr{X}_i}$ is the indicator function of $\mathscr{X}_i$, taking value 1 if $x \in \mathscr{X}_i$ and 0 otherwise.

Since the usual Euclidean topology on $\{0,1\}^n$ coincides with the discrete topology on $\{0,1\}^n$ and since a continuous functions to a discrete topological space are constant, see [49], then $\hat{h}$ only belongs to $C(\mathscr{X}, \{0,1\}^n)$ if it is trivial, i.e.: either $\mathscr{X}_i = \mathscr{X}$ or $\mathscr{X}_i = \emptyset$ for each $i$. Moreover, a direct computation shows that there are exactly $2^n$ functions in $C(\mathscr{X}, \{0,1\}^n)$. Thus, other topologies must be considered on $\{0,1\}^n$ in order to have a meaningful deterministic classification theory.

When $n = 1$, there are two other choices of topologies on $\{0,1\}$, up to homeomorphism. These are the trivial topology $\{\emptyset, \{0,1\}\}$ and the *Sierpiński topology* $\{\emptyset, \{1\}, \{0,1\}\}$. The trivial topology is uninteresting since a direct computation shows that with it every function in $C(\mathscr{X}, \{0,1\})$ becomes indistinguishable, i.e.: the co-topology on $C(\mathscr{X}, \{0,1\})$ becomes trivial and therefore density in $C(\mathscr{X}, \{0,1\})$ holds trivially for any non-empty subset. In the case of the Sierpiński topology in [53, Chapter 7] it is shown that all indicator functions of any open set $\mathscr{X}$ from any sufficiently regular topological space, such as $\mathscr{X}$, is a continuous function to $\{0,1\}$ with the Sherpiński topology. This latter property has lead to widespread use of this space in semantics.

The next result shows that $\hat{h}$ can be approximated on two fronts simultaneously. First, by showing that $\hat{h}$ has a natural decomposition as $I_{(\frac{1}{2},1]}$ applied component-wise to continuous *soft (fuzzy) classifier* $\hat{s}$, i.e. $\hat{s} \in C(\mathscr{X}, [0,1]^n)$, satisfying

$$\hat{s}_i^{-1}[(1/2,1]] = \mathscr{X}_i, \qquad (\forall i = 1, \ldots, n). \tag{7}$$

Subsequently, the architecture $\mathscr{F}_{\rho,\phi}$ is shown to simultaneously approximate $\hat{s}$ uniformly on compacts in $C(\mathscr{X}, [0,1]^n)$ and $\hat{h}$ in the compact-open topology on $C(\mathscr{X}, \{0,1\}^n)$. Intuitively, (7) represents the philosophy of logistic regression where one approximates on the interval and the thresholds the logistic classifier to obtain a strict decision rule, and thus a hard classifier.

**Theorem 3.11** (Universal Classification: General Case). *Let $\{0,1\}^n$ be equipped with the n-fold product of the Sierpiński topology on $\{0,1\}$, $\phi$ be continuous and injective, $\rho : \mathbb{R}^n \to (0,1)^n$ be a homeomorphism, $\alpha \in (0,1)$, and $\mathscr{F} \subseteq C(\mathbb{R}^m, \mathbb{R}^n)$ be dense. Let $\{\mathscr{X}_i\}_{i=1}^n$ be a set of open subsets of a metric space $\mathscr{X}$ and let $\hat{h}$ be its associated ideal classifier defined by (6). Then the following hold:*
  *(i) (Hard-Soft Decomposition) There exist continuous functions $\hat{s}_i \in C(\mathscr{X}, [0,1])$ such that*

$$\hat{h} = I_{(\alpha,1]} \bullet (\hat{s}_1, \ldots, \hat{s}_n) \qquad \hat{s}_i^{-1}[(\alpha,1]] = \mathscr{X}_i, (\forall i = 1, \ldots, n)$$

 *(ii) (Universal Classification) There exists a sequence $\{f_k\}_{k \in \mathbb{N}}$ in $\mathscr{F}$ such that:*
   *(a) (Soft Classification) For each non-empty compact subset $\kappa \subseteq \mathscr{X}$ and every $\varepsilon > 0$, there is some $K \in \mathbb{N}^+$ such that*

$$\sup_{x \in \kappa} \max_{i=1,\ldots,n} |\rho \circ f_k \circ \phi(x)_i - \hat{s}_i(x_i)| < \varepsilon, \qquad (\forall k \geq K)$$

   *(b) (Hard Classification) $I_{(\alpha,1]} \bullet \rho \circ f_k \circ \phi$ converges to $\hat{h}$ in $C(\mathscr{X}, \{0,1\}^n)$ for the co-topology.*

*Furthermore, $\mathscr{F}_{\rho,\phi}$ is dense in $C(\mathscr{X}, [0,1]^n)$.*

As an application, we now show that most feed-forward DNNs and deep CNNs used in practice for classifications, are indeed universal classifiers in the sense of Theorem 3.11.

Let $\sigma : \mathbb{R} \to \mathbb{R}$ be a continuous *activation* function, and let $\mathscr{N}\mathscr{N}^\sigma$ denote the set of feed-forward networks from $\mathbb{R}^m$ to $\mathbb{R}^n$, i.e.: continuous functions with representation

$$f(x) = W \circ f^{(J)}, \quad f^{(j)}(x) = \sigma \bullet \left( W^{(j)} \circ f^{(j-1)}(x) \right), f^{(0)}(x) = x, \quad j = 1, \ldots, J \tag{8}$$

where $W$ and $W^j$ are affine maps and $\bullet$ denotes component-wise composition. The following results directly follow from Theorem 3.11 and the central result of [35], and validates the principle way neural networks are used for classification.

**Corollary 3.12** (Universal Classification: Deep Feed-Forward Networks). *Let $\{\mathscr{X}_i\}_{i=1}^n$ be open subsets of $\mathscr{X}$, and $\hat{h}$ be their associated ideal classifier. Let $\phi : \mathscr{X} \to \mathbb{R}^n$ be a continuous injective feature map. Let $\sigma$ be a continuous, locally-bounded, and non-constant activation function. Let $\rho$ either be the component-wise logistic function. Then there exists a sequence $\{f_k\}_{k \in \mathbb{N}^+}$ of DNNs satisfying the conclusions of Theorem 3.11.*

Define the set of deep CNNs with ReLu activation and sparsity $2 \leq s \leq m$, denoted by $Conv^s$, to be the collection of all functions from $\mathbb{R}^n$ to $\mathbb{R}$ represented by

$$f(x) = W \circ f^{(J)}, \quad f^{(j)}(x) = \sigma \bullet \left( w^{(j)} \star (f^{(j-1)}(x)) - b^j \right), f^{(0)}(x) = x, \quad j = 1, \ldots, J,$$

where $W$ is an affine map from $\mathbb{R}^{d+Js}$ to $\mathbb{R}$, $b^{(j)} \in \mathbb{R}^{d+js}$, $w^{(j)} = \{w_k^{(j)}\}_{k=-\infty}^{\infty}$ is a *convolutional filter mask* where $w_k \in \mathbb{R}$ and $w_k \neq 0$ only if $0 \leq k \leq s$, and the *convolutional operation* of $w^{(j)}$ with the vectors $\{v_j\}_{j=1}^J$ is the sequence defined by $(w \star v)_i = \sum_{j=0}^{J-1} w_{i-j} v_j$ and $\sigma(x) = \max\{0, x\}$.

**Corollary 3.13** (Universal Classification: Deep CNNs). *Let $2 \leq s \leq n$, $\{\mathscr{X}_i\}_{i=1}^n$ be open subsets of $\mathscr{X}$, and $\hat{h}$ be their associated ideal classifier. Let $\phi : \mathscr{X} \to \mathbb{R}^n$ be a continuous injective feature map and let $\rho : \mathbb{R} \to (0, 1)$ be the logistic function. Then there is a sequence of deep CNNS $\{f_k\}_{k \in \mathbb{N}^+}$ in $Conv_{\rho, \phi}^s$ satisfying the conclusion of Theorem 3.11.*

## 3.2 Applications in Geometric Deep Learning

This subsection illustrates the applicability of the main results to geometric deep learning. Our examples focus on two illustrative points, first that many commonly used non-Euclidean regression models can be extended to non-Euclidean architectures capable of universal approximation and second, we illustrate how our results can be used to validate the approximation capabilities of certain geometric deep learning architectures.

For Cartan-Hadamard manifolds, the Cartan-Hadamard Theorem, [30, Corollary 6.9.1], guarantees that $\partial C_p = \emptyset$ and in particular $\text{Log}_p^{\mathscr{M}}$ is a globally-defined homeomorphism between $\mathscr{M}$ and $\mathbb{R}^m$. Thus, the following result follows from Corollary 3.8.

**Corollary 3.14** (Cartan-Hadamard Version). *Let $\mathscr{F}$ be dense in $C(\mathbb{R}^m, \mathbb{R}^n)$, let $\mathscr{M}$ and $\mathscr{N}$ be Cartan-Hadamard manifolds of dimension $m$ and $n$. Then, $\mathscr{F}_{\text{Log}_p^{\mathscr{M}}, \text{Exp}_q^{\mathscr{N}}}$ is dense in $C(\mathscr{M}, \mathscr{N})$.*

We consider here two consequences of this result.

### 3.2.1 Symmetric Positive-Definite Matrix Learning

Symmetric positive-definite matrices play a prominent role in many applied sciences, largely due to their relationship to covariance matrices, in areas ranging from computational anatomy in [47], computer vision in [46], and in finance [5]. The space $P_d^+$ of $d \times d$ symmetric positive-definite matrices is a non-Euclidean subspace of $\mathbb{R}^{d^2}$. In [1], $P_d^+$ is shown to be a Cartan-Hadamard manifold whose Riemannian exponential and logarithm maps are, respectively, given by

$$\text{Exp}_A(B) = \sqrt{A} \exp \left( \sqrt{A}^{-1} B \sqrt{A}^{-1} \right) \sqrt{A}, \text{Log}_A(B) = \sqrt{A} \log \left( \sqrt{A}^{-1} B \sqrt{A}^{-1} \right) \sqrt{A}, \quad (9)$$

where exp and log denote the matrix exponential and logarithms, respectively. Moreover, the distance function on this space is given by

$$d_+(A, B) \triangleq \left\| \sqrt{A} \log \left( \sqrt{A}^{-1} B \sqrt{A}^{-1} \right) \sqrt{A} \right\|_F,$$

where $\| \cdot \|_F$ denotes the Fröbenius norm and $\sqrt{A}$ is well-defined for any matrix in $P_d^+$. Using this distance, [40] developed non-Euclidean least-squares regression on $P_d^+$. The parameters involved in these models are typically optimized either using the non-Euclidean line search algorithms of [40] or the non-Euclidean stochastic gradient approach on $P_d^+$ of [6]. The aforementioned regression models can be extended to form a ucc-dense architecture in $C(P_d^+, P_D^+)$.

**Corollary 3.15** (Universal Approximation for Symmetric Positive-Definite Matrices). *Let $d, D \in \mathbb{N}^+$ and $\mathscr{F} \subseteq C(\mathbb{R}^{d(d+1)/2}, \mathbb{R}^{D(D+1)/2})$ be ucc-dense. Then, for any $A \in P_d^+$ and $B \in P_D^+$, $\mathscr{F}_{\text{Log}_A, \text{Exp}_B}$ is ucc-dense in $C(P_d^+, P_D^+)$. In particular, if $\sigma$ is a continuous, locally-bounded, and non-polynomial activation function then $\mathscr{N}\mathscr{N}^\sigma_{\text{Log}_A, \text{Exp}_B}$ is ucc-dense in $C(P_d^+, P_D^+)$.*

### 3.2.2 Hyperbolic Feed-Forward Networks

For $c > 0$, the *generalized hyperbolic spaces* $\mathbb{D}_c^n$ of [17] have underlying set $\{x \in \mathbb{R}^n : c\|x\|^2 < 1\}$ and their topology is induced by the following non-Euclidean metric

$$d_c(x, y) \triangleq \frac{2}{\sqrt{c}} \tanh^{-1} \left( \sqrt{c} \left\| \frac{(1 - c\|x\|^2)y - (1 - 2cx^\top y + c\|y\|^2)}{1 - 2cx^\top y + c^2\|x\|^2\|y\|^2} \right\| \right).$$

Though a direct description of hyperbolic feed-forward neural networks would be lengthy, on [17, page 6], it is shown any hyperbolic feed-forward network from $\mathbb{D}_c^m$ to $\mathbb{D}_c^n$ can be represented as

$$\left\{ \text{Exp}_0^{\mathbb{D}_c^k} \circ f \circ \text{Log}_0^{\mathbb{D}_c^k} : f \in \mathscr{N}\mathscr{N}^\sigma \right\}, \quad (10)$$

where $\mathrm{Exp}_0^{\mathbb{D}_c^k}$ is the Riemmanian Exponential map on $\mathbb{D}_c^k$ about 0, as in Corollary 3.14. Closed-form expressions are obtained in [17, Lemma 2] for these feature and readout maps. Since, as discussed in [17], $\mathbb{D}_c^k$ is a complete connected Riemannian manifold of non-positive curvature then the Cartan-Hadamard Theorem implies that $C_0 = \emptyset$. Whence, Corollary 3.14 yields the following.

**Corollary 3.16** (Hyperbolic Neural Networks are Universal)**.** *Let $\sigma$ be a continuous, non-polynomial, locally-bounded activation function and $c > 0$. Then for every $g \in C(\mathbb{D}_c^m, \mathbb{D}_c^n)$, every $\varepsilon > 0$, and every compact subset $K \subseteq \mathbb{D}_c^m$ there exists a hyperbolic neural network $g_{\varepsilon,K,c}$ in (10) satisfying*

$$\sup_{x \in K} d_c(g(x), g_{\varepsilon,K,c}) < \varepsilon.$$

Next, applications of Theorems 3.3 and 3.10 with Euclidean input and output spaces are considered.

## 3.3 Universality of Deep Networks with First Layers Randomized

Fix $\mathbb{R}$-valued random variables $\{X_i\}_{i=1}^k$ and $\{Z_i\}_{i=1}^k$ defined on a common probability space $(\Omega, \Sigma, \mathbb{P})$. Fix an activation function $\sigma : \mathbb{R} \to [0,1]$, and positive integers $\{d_i\}_{i=1}^k$. Using this data, for each $i = 1, \ldots, k$ define random affine maps $W_i : \mathbb{R}^{d_i} \times \Omega \to \mathbb{R}^{d_{i+1}}$, defined by

$$(x, \omega) \mapsto A_i(\omega)x + b_i(\omega), \tag{11}$$

where the entries of $A_i$ are i.i.d. copies of $X_i$ and the entries of $b_i$ are i.i.d. copies of $Z_i$.

The random affine maps (11) define the (random) set of deep feed-forward neural networks with first $k$ layers randomized and last 2 layers trainable to be the (random) subset of $C(\mathbb{R}^d, \mathbb{R}^D)$ via

$$\mathscr{N}\!\mathscr{N}_{2,k}^\sigma(\omega) \triangleq \{f \in C(\mathbb{R}^m, \mathbb{R}^n) : (\exists g \in \mathscr{N}\!\mathscr{N}_2^\sigma) f(x) = g \circ [\sigma \bullet W_k(x, \omega) \circ \sigma \bullet \cdots \circ \sigma \bullet W_1(x, \omega)]\},$$

where $\mathscr{N}\!\mathscr{N}_2^\sigma$ is the collection of feed-forward neural networks of the form $W_2 \circ \sigma \bullet W_1$, where $W_1 : \mathbb{R}^m \to \mathbb{R}^d$ and $W_2 : \mathbb{R}^d \to \mathbb{R}^n$ are affine maps and $d$ is a positive integer. Under the following mild assumptions, the random set $\mathscr{N}\!\mathscr{N}_{2,k}^\sigma$ is dense in $C(\mathbb{R}^m, \mathbb{R}^n)$ with probability 1.

**Assumption 3.17.** *For each $i = 1, \ldots, k$*
  *(i) $d_i \le d_{i+1}$ for each $i = 1, \ldots, k$,*
  *(ii) $\sigma$ is a strictly increasing and continuous,*
  *(iii) $\mathbb{E}[X_i] = \mathbb{E}[Z_i] = 0$, $\mathbb{E}[X_i^2] = \mathbb{E}[Z_i^2] = 1$,*
  *(iv) For every $C > 1$, $\mathbb{E}[|X_i|^C], \mathbb{E}[|Z_i|^C] < \infty$.*

**Theorem 3.18.** *If Assumption 3.17 holds, then there exists a measurable subset $\Omega' \subseteq \left\{ \omega \in \Omega : \overline{\mathscr{N}\!\mathscr{N}_{2,k}^\sigma(\omega)} = C(\mathbb{R}^d, \mathbb{R}^D) \right\}$ satisfying $\mathbb{P}(\Omega') = 1$.*

**Corollary 3.19** (Sub-Gaussian Case with Sigmoid Activation)**.** *Let $X_i = Z_i$ for each $i = 1, \ldots, k$ be independent standardized sub-Gaussian random-variables, $\sigma(x) = \frac{1}{1+e^{-x}}$, and $d_i = d$ for each $i = 1, \ldots, k$. Then the conclusion of Theorem 3.18 holds.*

**Corollary 3.20** (Bernoulli Case with PReLU Activation)**.** *Suppose that for every $i, j = 1, \ldots, k$, $X_i$ and $Z_j$ i.i.d. copies of a random variable taking values $\{-1, 1\}$ with probabilities $\{\frac{1}{2}, \frac{1}{2}\}$. Let $d_i = d$ for each $i = 1, \ldots, k$ and $\sigma$ be the PReLU activation function of [22]. Then Assumptions 3.17 are met; thus the conclusion of Theorem 3.18 holds.*

## 3.4 Feed-Forward Layers of Sub-Minimal Width

In this section, we use Theorem 3.10 to describe how additional layers can be incorporated into a DNN, which violate the minimum width requirements of $m + 1$ in its hidden layers (see [29, 45]) but do not negatively impact the architecture's approximation capabilities. We say that such layers have *sub-minimal width*. We derive specific conditions on the activation functions and structure of the connections between sub-minimal width layer ensuring that Assumptions 3.1 and 3.2 are met.

**Proposition 3.21** (Input Layers of Sub-Minimal Width: Continuous Monotone Activations and Invertible Connections)**.** *Let $\sigma$ be a continuous and strictly increasing activation function, $J \in \mathbb{N}_+$, $A_1, \ldots, A_J$ be $m \times m$ matrices, and $b_1, \ldots, b_J \in \mathbb{R}^d$. Let $\phi(x) \triangleq \phi_J(x)$ where*

$$\phi_j(x) \triangleq \sigma \bullet \left( \exp(A_j) \phi_{j-1}(x) + b_j \right) \qquad \phi_0(x) \triangleq x, \quad j = 1, \ldots, J, \tag{12}$$

*where* exp *is the matrix exponential. Then $\phi$ satisfies Assumption 3.1.*

**Proposition 3.22** (Output Layers of Sub-Minimal Width: Invertible Feed-Forward Layers)**.** *In the setting of Proposition 3.21, if $\sigma$ is also surjective then $\phi$ is a homeomorphism, and in particular it satisfies Assumption 3.2.*

**Example 3.23** (Sub-Minimal Width via Generalized PReLU Activation). *Fix $\alpha, \beta \in (0, \infty)$, $\alpha \neq \beta$. Then the activation function $\sigma = \beta x I_{x \geq 0} + \alpha x I_{x < 0}$ is monotone increasing and surjective.*

**Example 3.24** (ReLU Feed-Forward Layers Cannot Achieve Sub-Minimal Width). *Let $\phi$ be as in (12) and let $\sigma(x) \triangleq \max\{0, x\}$. By Theorem 3.4, $\mathscr{F}_{1_{\mathbb{R}^n}, \phi}$ is not dense in $C(\mathbb{R}^m, \mathbb{R}^n)$.*

### 3.4.1 Numerical Illustration

The following numerical illustration will also make use of the following method for constructing feature maps satisfying Assumption 3.1. The method can be interpreted as passing the inputs through an arbitrary function and feeding them into the model's input via a skip connection.

**Proposition 3.25** (Skip Connections Using Arbitrary Continuous Functions are Good Feature Maps). *Let $d \in \mathbb{N}_+$ and $g \in C(\mathbb{R}^m, \mathbb{R}^d)$. Then $\phi_g(x) \triangleq (x, g(x))$ satisfies Assumption 3.1.*

**Example 3.26** (Pre-Trained DNN with a Skip Connection are Good Feature Maps). *Let $g \in \mathcal{NN}^\sigma$. Then $\phi_g(x) \triangleq (x, g(x))$ satisfies Assumption 3.1.*

To illustrate the effect of (in)correctly choosing the networks' input and output layers we implement different DNNs whose initial or final layers are build using the above examples or intentionally fail Assumptions 3.1 or 3.2. Our implementations are on the California housing dataset [31], with the objective of predicting the median housing value given a set of economic and geo-spacial factors as described in [18]. The test-set consists of 30% percent of the total 20k training instances. The implemented networks are of the form $\rho \circ f \circ \phi$, where $f = W_2 \circ \sigma \bullet W_1$ is a shallow feed-forward network with ReLU activation and $\rho, \phi$ are built using the above examples.

Our reference model (Vanilla) is simply the shallow feed-forward network $f$. For the first DNN, which we denote (Bad), $\rho$ and $\phi$ are given by as in Example 3.24 and therefore violate Assumption 3.1. For the second DNN, denoted by (Good), $\rho$ and $\phi$ are as in Example (3.23) and Assumptions 3.1 and 3.2 are met. The final DNN, denoted by (Rand), $\rho$ is as in Example 3.23 and $\phi$ is as in Example 3.26 where the pre-trained network is generated randomly following in Corollary 3.20.

| Model | Test | | | | Train | | | |
|---|---|---|---|---|---|---|---|---|
| | Good | Rand | Bad | Vanilla | Good | Rand | Bad | Vanilla |
| MAE | 0.318 | 0.320 | 0.876 | 0.320 | 0.252 | 0.296 | 0.887 | 0.284 |
| MSE | 0.247 | 0.259 | 1.355 | 0.257 | 0.174 | 0.234 | 1.409 | 0.209 |
| MAPE | 16.714 | 17.626 | 48.051 | 17.427 | 12.921 | 15.668 | 48.698 | 14.878 |

Table 1: Training and test predictive performance.

As anticipated, Table 1 shows that selecting the networks' initial and final layers according to our method either improves performance (Good) when all involved parameters are trainable or does not significantly affect it even if nearly every parameter is random (Rand). However, disregarding Assumptions 3.1 and 3.2 when adding additional deep layers dramatically degrades predictive performance, as is the case for (Bad). Table 1 shows that if a DNN's first and final layers are structured according to Theorem 3.10 then expressibility can be improved, even if these layers violate the minimum width bounds of [29, 45]. Python code for these implementations is available at [33].

## 4 Conclusion

Modifications to the input and output layers of any neural networks, using carefully chosen feature $\phi : \mathscr{X} \to \mathbb{R}^m$ and readout $\rho : \mathbb{R}^n \to \mathscr{Y}$ maps, are common in practice. Theorems 3.3, 3.10, and Corollary 3.14 provided general conditions on these maps guaranteeing that the new, modified, architecture can approximate any function in the uniform convergence on compacts (or more generally the compact-open) topologies between their new input and output spaces.

As a consequence of our main results, we showed that universal approximation implies universal classification once a component-wise logistic map is applied. This is a deterministic strengthening of the probabilistic results of [15]. We derived a method for constructing universal approximators between a wide class of manifolds. In particular, we extended the symmetric positive-definite matrix-valued regressor of [40] to a universal approximator and we showed that the hyperbolic feed-forward networks of [17] are universal approximators between hyperbolic spaces.

Our main results also described how to structure the first and final layers of a DNN between Euclidean spaces, so as to preserve the approximation capabilities of the network's middle layers. In particular, we provided conditions on a network's activation function and connections so that these layers can be made narrower than the specifications of [29, 45] while maintaining the architecture's expressibility. Lastly, we showed that randomly generated DNNs are good feature maps with probability 1.

## Broader Impact

A large portion of available data is non-Euclidean, either in the form of social network data to imaging data relevant in health applications of deep learning. The tools in this paper open up a generic means of translating the currently available deep learning technology to those milieus. The automation of tools in the medical sciences is important to helping reducing waiting times in hospitals and help make healthcare more accessible to all, so in that way, any automatizing of health science tools helps move society in that direction. Therefore, we hope that the methods presented in paper form a small step towards a greater positive advancement of the social and natural sciences.

## Acknowledgments and Disclosure of Funding

The authors would like to thank Gaël Meigniez for his insightful surrounding the differential topology tools used in the proof of Theorem 3.3. The authors would also like to thank Josef Teichmann, Behnoosh Zamanlooy, and Philippe Casgrain for their feedback and suggestions. This research was funded by the ETH Zürich foundation. The authors are grateful for its financial support of the project.

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
