[Supplementary Material]

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

This supplement provides theoretical justification of the claims made in this paper. We repeat the assumptions needed for our results and we repeat the statement of each theorem before it's proof for a smoother read.

## A   Assumptions

**Assumption** (Feature Map Regularity). *The map $\phi$ is a continuous and injective.*

**Assumption** (Readout Map Regularity). *Suppose that the readout map $\rho$ satisfies the following:*
  (i) *Either of the following hold:*
      (a) *$\rho$ is a continuous and it has a section on $\mathrm{Im}(\rho)$,*
      (b) *$\rho$ is a covering projection of $\mathbb{R}^m$ onto $\mathrm{Im}(\rho)$ and $\mathscr{X}$ is connected and simply connected,*
  (ii) *$\mathrm{Im}(\rho)$ is dense in $\mathscr{Y}$,*
  (iii) *$\partial\mathrm{Im}(\rho)$ is collared; i.e.: there exists an open subset $U \subseteq \mathscr{Y}$ containing $\partial\mathrm{Im}(\rho)$ and a homeomorphism $\psi : U \to \partial\mathrm{Im}(\rho) \times [0,1)$ mapping $\partial\mathrm{Im}(\rho)$ to $\partial\mathrm{Im}(\rho) \times \{0\}$.*

**Assumption** (Readout Map Regularity: Geometric Version). *Suppose that $\rho$ satisfies:*
  (i) *$\rho$ satisfies Assumption 3.2 (i) and $\mathrm{Im}(\rho) \subseteq \mathrm{Int}(\mathscr{Y})$,*
  (ii) *$\mathrm{Int}(\mathscr{Y}) - \mathrm{Im}(\rho)$ is a (possibly empty) smooth submanifold of $\mathrm{Int}(\mathscr{Y})$ of dimension strictly less-than $\dim(\mathrm{Int}(\mathscr{Y})) - n$.*

**Assumption** (Regularity of Randomized Deep Networks). *For each $i = 1,\dots,k$*
  (i) *$d_i \le d_{i+1}$ for each $i = 1,\dots,k$,*
  (ii) *$\sigma$ is a strictly increasing and continuous,*
  (iii) *$\mathbb{E}[X_i] = \mathbb{E}[Z_i] = 0$, $\mathbb{E}[X_i^2] = \mathbb{E}[Z_i^2] = 1$,*
  (iv) *For every $C > 1$, $\mathbb{E}[|X_i|^C], \mathbb{E}[|Z_i|^C] < \infty$.*

## B   Proofs

### B.1   Proof of Main Results

In the next lemma, we use the term algebra in the sense of the Stone-Weirestrass theorem, see [12, V.8] for details.

**Lemma B.1.** *Let $\phi : X \to Z$ be a continuous injection between topological spaces. Let $\mathscr{F} \subset C(Z, \mathbb{R}^n)$ be dense in the compact-open topology. Then $\{f \circ \phi \mid f \in \mathscr{F}$ is dense in $C(X, \mathbb{R}^n)$.*

*Proof.* Let $\Phi : C(Z, \mathbb{R}^n) \to C(X, \mathbb{R}^n)$ be defined by $[\Phi(f)](x) = f(\phi(x))$, which by [43, Theorem 46.11] is a continuous map. Hence, $\overline{\Phi(\mathscr{F})} = \overline{\Phi(C(Z, \mathbb{R}^n))}$, because $\mathscr{F}$ is dense. Therefore, in order to show that $\Phi(\mathscr{F})$ is dense, it is enough to prove density of $\Phi(C(Z, \mathbb{R}^n))$.

---
[*]Department of Mathematics, Eidgenössische Technische Hochschule Zürich, HG G 32.3, Rämistrasse 101, 8092 ürich, Switzerland. email: *anastasis.kratsios@math.ethz.ch*

[†]Department of Mathematics and Statistical Sciences, University of Alberta, 11324 89 Ave NW, Edmonton, AB T6G 2J5, Canada. email: *bilokopy@ualberta.ca*

Observe that $C(X, \mathbb{R}^n) = C(X) \oplus ... \oplus C(X)$, $C(Z, \mathbb{R}^n) = C(Z) \oplus ... \oplus C(Z)$ and $\Phi : C(Z) \to C(X)$ on every component independently. Hence, we just need to show that $\Phi(C(Z))$ is dense in $C(X)$.

To that end, observe that since $\Phi$ is an algebra homomorphism, $\Phi(C(Z))$ is a subalgebra of $C(X)$. Clearly, $1 \in \Phi(C(Z))$, and since $\phi$ is an injection, it is easy to see that $\Phi(C(Z))$ separates points of $X$. Thus, $\Phi(C(Z))$ is dense by the Stone-Weierstrass theorem. $\qquad\square$

**Lemma B.2.** *Let $\mathscr{X}$ be a topological space, $\rho : \mathbb{R}^n \to \mathscr{Y}$ be a map satisfying Assumption 3.2, and $\mathscr{F}$ be dense in $C(X, \mathbb{R}^n)$ for the compact-open topology. Then the set*

$$\{ g \in C(\mathscr{X}, \mathrm{Im}(\rho)) : (\exists f \in \mathscr{F}) \, g = \rho \circ f \}, \tag{13}$$

*is dense in $C(\mathscr{X}, \mathrm{Im}(\rho))$. In particular, if $\mathscr{Y}$ is a metric space then the set (13) is dense in $C(\mathscr{X}, \mathrm{Im}(\rho))$ for the ucc topology.*

*Proof.* First, suppose that Assumption 3.2 (i.a), (ii), and (iii) hold and that $\rho$ is continuous with continuous section. Define the map $F : C(\phi(\mathscr{X}), \mathbb{R}^n) \to C(\mathscr{X}, \mathrm{Int}(\rho))$ by $f \to \rho \circ f$. Since $\rho$ is continuous then by [43, Theorem 46.11] $F$ is also continuous. Moreover, by Assumption 3.2 (i.a), since there is a section $R$ to $\rho$, i.e. a continuous map $R : \mathrm{Int}(\rho) \to \mathbb{R}^n$ such that $\rho \circ R = 1_{\mathscr{Y}}$ then by [43, Theorem 46.11] the map $G : C(\mathscr{X}, \mathrm{Int}(\rho)) \to C(\mathscr{X}, \mathbb{R}^n)$ defined by $g \to R \circ g$ is well-defined and continuous. Furthermore, for every $g \in C(\mathscr{X}, \mathrm{Int}(\rho))$ it follows that

$$
\begin{aligned}
F \circ G(g) &= F(R \circ g) \\
&= \rho \circ (R \circ g) \\
&= (\rho \circ R) \circ g \\
&= g \\
&= 1_{\mathrm{Int}(\rho)}(g),
\end{aligned}
$$

therefore $F$ is a continuous surjection. Since continuous surjections map dense sets to dense sets in their codomain, then since $\mathscr{F}$ is dense in $C(\mathscr{X}, \mathbb{R}^n)$ and the image of $\mathscr{F}$ under $F$ is the set of (13) then the set described in (13) is dense in $C(\mathscr{X}, \mathrm{Int}(\rho))$.

Now, suppose that Assumption 3.2 (i.b), (ii), and (iii) hold. Since $\mathscr{X}$ is simply connected and $\rho$ is a covering map, then the conditions for [52, Chapter 2, Section 2, Theorem 5] are met (since simply connectedness means a trivial fundamental group), therefore, for every $f \in C(\mathscr{X}, \mathrm{Im}(\rho))$ there exists some $\hat{f} \in C(\mathscr{X}, \mathbb{R}^n)$ such that

$$\rho \circ \hat{f} = f. \tag{14}$$

Since $\mathscr{F}$ is dense in $C(\mathscr{X}, \mathbb{R}^n)$ then exists a sequence $\{f_k\}_{k \in \mathbb{N}}$ in $\mathscr{F}$ converging to $\hat{f}$ in the compact-open topology. Since $\rho$ is continuous the continuity of $F$, following the same notation as the first part of the proof, implies that $\{F(f_k)\}_{k \in \mathbb{N}}$ converges to $F(\hat{f})$ in the compact-open topology. By (14) this implies that $\{F(f_k)\}_{k \in \mathbb{N}} = \{\rho \circ f_k\}_{k \in \mathbb{N}}$ converges to

$$f = F(f) = f$$

in the compact-open topology. Therefore, (13) is dense in $C(\mathscr{X}, \mathrm{Im}(\rho))$.

In particular, if $\mathscr{Y}$ is a metric space, then $\mathrm{Im}(\rho)$ is a metric space. Thus, by [43, Theorem 46.8] the compact-open and ucc topologies coincide on $C(\mathscr{X}, \mathrm{Im}(\rho))$. This gives the final claim. $\qquad\square$

It is convenient to sumamrize the above lemmas into a single lemma.

**Lemma B.3.** *If Assumptions 3.1 and 3.2 both hold, and if $\mathscr{F}$ is dense in $C(\mathbb{R}^m, \mathbb{R}^n)$ then $\mathscr{F}_{\rho,\phi}$ is dense in $C(\mathscr{X}, \mathrm{Im}(\rho))$ for the compact-open topology. Moreover, if $\mathscr{Y}$ is a metric space then $\mathscr{F}_{\rho,\phi}$ is dense in $C(\mathscr{X}, \mathrm{Im}(\rho))$ for the ucc-topology.*

*Proof.* The result follows directly from Lemmas B.2 and B.1. $\qquad\square$

**Lemma B.4.** *If $\mathscr{X}$ is locally-compact and Assumption 3.2 holds then $C(\mathscr{X}, \mathrm{Im}(\rho))$ is dense in $C(\mathscr{X}, \mathscr{Y})$ in the compact-open topology.*

*Proof of Lemma B.4.* By Assumption 3.2 (iii) there is a homeomorphism $\psi : U \to \partial \text{Im}(\rho) \times [0,1)$ and a (continuous) inclusion map $i : U \to \mathscr{Y}$. Therefore, the [14, Lemma 5.29.1] (existence and universal property of colimits of topological spaces and the fact that any push-out is a colimit) there exists a unique topological space (up to homeomorphism) $\mathscr{Y}'$ containing $\partial \text{Im}(\rho) \times [0,1)$ as a subspace for which there exists a continuous $\Psi : \mathscr{Y} \to \mathscr{Y}'$ satisfying

$$\Psi \circ i = j \circ \psi, \tag{15}$$

where $j$ is the (continuous) inclusion map of $\partial \text{Im}(\rho) \times [0,1)$ into $\mathscr{Y}'$, such that for any other topological space $\mathscr{Y}''$ satisfying (15) there exists a continuous map $I : \mathscr{Y}' \to \mathscr{Y}''$. In particular, this means that since $\psi$ is a homeomorphism then $\Psi$ must also be a homeomorphism.

Therefore, $C(\mathscr{X}, \Psi(\text{Im}(\rho)))$ is dense in $C(\mathscr{X}, \mathscr{Y}')$ if and only if $C(\mathscr{X}, \text{Im}(\rho))$ is dense in $C(\mathscr{X}, \mathscr{Y})$. Thus, without loss of generality we establish the result in the former setting where we have identified $U$ with $\partial \text{Im}(\rho) \times [0,1)$. When required by the context, we use the notation $x = (u^x, t^x) \in \partial \text{Im}(\rho) \times [0,1) = \Psi[U]$.

Accordingly, for any fixed $f \in C(\mathscr{X}, \mathscr{Y}')$ we define a sequence $\{f_n\}_{n \in \mathbb{N}}$ in $C(\mathscr{X}, \Psi(\text{Im}(\rho)))$ converging to $f$. For every $s \in (0,1)$ define the continuous functions $\tilde{\psi}_s : \partial \text{Im}(\rho) \times [0,1) \to \partial \text{Im}(\rho) \times (0,1)$ by

$$\tilde{\psi}_s(u,t) \triangleq \begin{cases} (u,t) & : t > s \\ (u,s) & : t \leq s. \end{cases}$$

Using the family $\{\tilde{\psi}_s\}_{s \in (0,1)}$ we define the sequence of functions $\{\psi_n\}_{n \in \mathbb{N}}$ from $\mathscr{Y}$ to $\text{Im}(\rho)$ by

$$\psi_n(x) \triangleq \begin{cases} x & : x \notin U \\ (u^x, \frac{1}{n}) & : x \in U \text{ and } t^x \leq \frac{1}{n}. \end{cases}$$

Finally, for every $n \in \mathbb{N}$, we set $f_n \triangleq \psi_n \circ f$. Note that, for each $n \in \mathscr{N}$, since $\psi(\partial \text{Im}(\rho)) = \text{Im}(\rho) \times \{0\}$ then $f_n \in C(\mathscr{X}, \Psi(\text{Im}(\rho)))$. Note also that, by construction $f_n(x) = f(x)$ for every $x \notin \tilde{X}$, where $\tilde{X} \triangleq f^{-1}[\partial \text{Im}(\rho) \times [0,1)]$, and both $f_n$ and $f$ map $\tilde{X}$ to $\partial \text{Im}(\rho) \times [0,1)$. Thus, it is enough to show that $f_n$ converges to $f$ in $C(\tilde{X}, \partial \text{Im}(\rho) \times [0,1))$.

Since $\phi$ is a continuous injection into the locally-compact space $\mathbb{R}^m$ then $\mathscr{X}$, and therefore $\tilde{X}$, are locally-compact. Therefore, $C(\tilde{X}, \partial \text{Im}(\rho) \times [0,1))$ is homeomorphic to $C(\tilde{X}, \partial \text{Im}(\rho)) \times C(\tilde{X}, [0,1))$. By [43, Section 19.2, Exercise 6], $f_n$ converges to $f$ on the product space $C(\tilde{X}, \partial \text{Im}(\rho)) \times C(\tilde{X}, [0,1))$ if and only if $p_i(f_n)$ converges to $p_i(f)$ for $i = 1, 2$ where $p_1$ is the canonical projection of $C(\tilde{X}, \partial \text{Im}(\rho)) \times C(\tilde{X}, [0,1))$ onto $C(\tilde{X}, \partial \text{Im}(\rho))$ and $p_2$ is the canonical projection of $C(\tilde{X}, \partial \text{Im}(\rho)) \times C(\tilde{X}, [0,1))$ onto $C(\tilde{X}, [0,1))$. First observed that,

$$p_1(f_n) = f = p_1(f),$$

for each $n \in \mathbb{N}$. Next,

$$p_2(f_n) = \frac{1}{n} I_{[0,\frac{1}{n}]} + t I_{[\frac{1}{n},\infty)} \text{ and } p_2(f) = t.$$

Since $[0,1)$ is topologized with the (relative) Euclidean topology which is metric then [43, Theorem 46.8] implies that compact-open topology on $C(\tilde{X}, [0,1))$ agrees with the topology of uniform convergence on compacts in $C(\tilde{X}, [0,1))$. Fix $\varepsilon > 0$ and $n \geq \frac{1}{\varepsilon}$. Thus, for every compact $\tilde{K} \subseteq \tilde{X}$

$$\sup_{x \in \tilde{K}} \|p_2(f_n)(x) - p_2(f)(x)\| = \max_{t \in [0,\frac{1}{n}]} \left| \frac{1}{n} - t \right| \leq \frac{1}{n} < \varepsilon.$$

Therefore, $p_2(f_n)$ converges to $p_2(f)$ in the compact-open topology on $C(\tilde{X}, [0,1))$. Hence, $f_n$ converges to $f$ in the compact-open topology on $C(\tilde{X}, \partial \text{Im}(\rho)) \times C(\tilde{X}, [0,1))$ and therefore on $C(\tilde{X}, \partial \text{Im}(\rho) \times [0,1))$. Since

$$\left\{ U_{\tilde{K},\tilde{O}} : \emptyset \neq \tilde{K} \subseteq \tilde{\mathscr{X}} \text{ compact and } \emptyset \neq \tilde{O} \subseteq \partial \text{Im}(\rho) \times [0,1) \text{ open} \right\}$$
$$U_{\tilde{K},\tilde{O}} \triangleq \left\{ f \in C(\tilde{\mathscr{X}}, \partial \text{Im}(\rho) \times [0,1)) : f(\tilde{K}) \subseteq \tilde{O} \right\} \tag{16}$$

is a sub-base for the compact-open topology on $C(\tilde{\mathscr{X}}, \partial \text{Im}(\rho) \times [0,1))$ then by definition of convergence, for every $\tilde{K}_1, \ldots, \tilde{K}_n \subseteq \tilde{\mathscr{X}}$ compact and $\tilde{O}_1, \ldots, \tilde{O}_n \subseteq \partial \text{Im}(\rho) \times [0,1)$ open if $f \in \bigcap_{i=1}^{n} U_{\tilde{K}_i, \tilde{O}_i}$ then there exists some $N \in \mathbb{N}$ for which $f_N \in \bigcap_{i=1}^{n} U_{\tilde{K}_i, \tilde{O}_i}$.

Since every compact subset $K \subseteq \mathscr{X}$ is relatively compact in $\tilde{X}$ and every open subset $O \subseteq \mathscr{Y}$ is relatively compact in $\partial \mathrm{Im}(\rho)) \times C(\tilde{X}, [0,1))$ then if $f \in \bigcap_{i=1}^{n} V_{K_i, O_i}$ where $V_{K_i, O_i}$ are as in (3) then

$$f(K_i) \subseteq O_i \qquad i = 1, \ldots, n.$$

Therefore, there exists some $N \in \mathbb{N}$ satisfying

$$f_N(K_i \cap \tilde{\mathscr{X}}) \subseteq O_i \cap \partial \mathrm{Im}(\rho) \times [0,1) \qquad i = 1, \ldots, n.$$

However, by construction, $f_N = f$ on $\mathscr{X} - \tilde{\mathscr{X}}$ and therefore on each $K_i - K_i \cap \mathscr{X}$. Therefore,

$$f_N(K_i) \subseteq O_i \qquad i = 1, \ldots, n.$$

Thus, $f_N \in \bigcap_{i=1}^{n} V_{K_i, O_i}$ and $f_n$ converge to $f$ in $C(\mathscr{X}, \mathscr{Y})$ for the compact-open topology. $\qquad \square$

We may now prove the following result and its consequences.

**Theorem** (General Version). *Suppose that $\mathscr{F}$ is dense in $C(\mathbb{R}^m, \mathbb{R}^n)$. If Assumptions 3.1 and 3.2 hold then $\mathscr{F}_{\rho, \phi}$ is dense in $C(\mathscr{X}, \mathscr{Y})$.*

*Proof.* Since Assumptions 3.1 and 3.2 hold and since $\mathscr{F}$ is dense in $C(\mathbb{R}^m, \mathbb{R}^n)$ then $\mathscr{F}_{\rho, \phi}$ is dense in $C(\mathscr{X}, \mathrm{Im}(\rho))$ according to Lemma (B.3). By Lemma B.4, $C(\mathscr{X}, \mathrm{Im}(\rho))$ is dense in $C(\mathscr{X}, \mathscr{Y})$. Since density is transitive, then $\mathscr{F}_{\rho, \phi}$ is dense in $C(\mathscr{X}, \mathscr{Y})$. $\qquad \square$

**Theorem** (Sharpness of Assumption 3.1). *Let $\mathscr{X}$ be a metrizable manifold with boundary, let $\phi$ be continuous, and $\mathscr{F} \subseteq C(\mathbb{R}^m, \mathbb{R}^n)$. Then $\mathscr{F}_{1_{\mathbb{R}^n}, \phi}$ is dense in $C(\mathscr{X}, \mathbb{R}^n)$ if and only if $\phi$ is injective.*

*Proof.* Let $\phi : \mathscr{X} \to \mathbb{R}^m$ be a continuous function.

Assume that $\mathscr{F}$ is dense in $C(\mathbb{R}^m, \mathbb{R}^n)$. Since $1_{\mathbb{R}^n}$ is a homeomorphism of $\mathbb{R}^n$ onto itself it satisfies Assumption 3.2 (i) and (ii). Since $\mathrm{Im}(1_{\mathbb{R}^n}) = \emptyset$ then Assumption 3.2 (iii) holds. Therefore, Theorem 3.3 implies that $\mathscr{F}_{1_{\mathbb{R}^n}, \phi}$ is dense in $C(\mathscr{X}, \mathbb{R}^n)$.

Conversely, assume that $\phi$ is not injective. Suppose that $\mathscr{F}_{1_{\mathbb{R}^n}, \phi}$ is dense in $C(\mathscr{X}, \mathbb{R}^n)$. Since $\phi$ is not injective, then there exists distinct $x_1^\star, x_2^\star \in \mathscr{X}$ such that

$$\phi(x_1^\star) \neq \phi(x_2^\star). \tag{17}$$

Since $\mathscr{X}$ is metrizable then [43, Theorem 32.2] implies that $\mathscr{X}$ is normal; i.e. : if $K_1, K_2 \subseteq \mathscr{X}$ are disjoint closed subsets then there exist disjoint open subsets $U_1, U_2 \subseteq \mathscr{X}$ such that $K_i \subseteq U_i$ for $i = 1, 2$. Since $\mathscr{X}$ is normal then [43, Urysohn's Lemma; Theorem 33.1] implies that $C(\mathscr{X}, \mathbb{R})$ separates points in $\mathscr{X}$; i.e.: for every distinct $x_1, x_2$ there exists some $\tilde{f}_{x_1, x_2} \in C(\mathscr{X}, \mathbb{R})$ such that

$$\tilde{f}_{x_1, x_2}(x_1) \neq \tilde{f}_{x_1, x_2}(x_2). \tag{18}$$

Since $\mathbb{R}$ is a metric subspace of $\mathbb{R}^m$ then the inclusion map $\iota : \mathbb{R} \to \mathbb{R}^m$ taking $x$ to $(x, 0, \ldots, 0)$ is continuous and by definition it is injective. Therefore, by (18) the function $f \triangleq \iota \circ \tilde{f}_{x_1^\star, x_2^\star} : \mathscr{X} \to \mathbb{R}^n$ satisfies

$$f(x_1) \neq f(x_2). \tag{19}$$

Since $\mathscr{F}_{1_{\mathbb{R}^n}, \phi}$ was assumed to be dense in $C(\mathscr{X}, \mathbb{R}^n)$ then there exists a sequence $\{f_k\}_{k \in \mathbb{N}}$ in $\mathscr{F}_{1_{\mathbb{R}^n}, \phi}$ converging to $f$ in $C(\mathscr{X}, \mathbb{R}^n)$. Since $\mathscr{X}$ is locally-compact then [43, Theorem 46.10] implies that the evaluation function $e : \mathscr{X} \times C(\mathscr{X}, \mathbb{R}^n)$ mapping $(x, g) \mapsto g(x)$ is continuous. Since $\mathbb{R}^m$ is a metric space then $C(\mathscr{X}, \mathbb{R}^m)$ is a metric and since continuity in metric spaces is equivalent to sequential continuity then in particular

$$\lim_{k \uparrow \infty} e(x_i^\star, f_k) = e\left(x_i^\star, \lim_{k \uparrow \infty} f_k\right) = e(x_i^\star, f) = f(x_i^\star), \tag{20}$$

for $i = 1, 2$. Since, for each $k \in \mathbb{N}$, $f_k \in \mathscr{F}_{1_{\mathbb{R}^n}, \phi}$ then there exists some $g_k \in C(\mathbb{R}^m, \mathbb{R}^n)$ satisfying $f_k = g_k \circ \phi$. Hence, (17) implies that, for each $k \in \mathbb{N}$, $f_k(x_1^\star) = g_k \circ \phi(x_1^\star) = g_k \circ \phi(x_2^\star) = f_k(x_2^\star)$; thus, (20) implies that

$$f(x_2^\star) = \lim_{k \uparrow \infty} e(x_2^\star, f_k) = e(x_2^\star, f_k) = f_k(x_2^\star) = f_k(x_1^\star) = e(x_1^\star, f_k) = \lim_{k \uparrow \infty} e(x_1^\star, f_k) = e(x_1^\star, f) = f(x_1^\star).$$

However, $f(x_1^\star) \neq f(x_2^\star)$ according to (19); a contradiction. Therefore, $\mathscr{F}_{1_{\mathbb{R}^n}, \phi}$ cannot be dense in $C(\mathscr{X}, \mathbb{R}^n)$. $\qquad \square$

**Corollary.** *If $\phi$ is a continuous injective map, $\rho$ is a surjective covering projection, and $\mathscr{F}$ is dense in $C(\mathbb{R}^d, \mathbb{R}^D)$ then $\mathscr{F}_{\rho,\phi}$ is dense in $C(\mathscr{X}, \mathscr{Y})$. In particular, $\phi$ and $\rho$ may be homeomorphisms.*

*Proof.* Since $\rho$ is a continuous surjection then $\mathrm{Int}\,(\rho)$ is trivially dense in $\mathscr{Y}$, since
$$\mathscr{Y} - \mathrm{Int}\,(\rho) = \mathscr{Y} - \mathscr{Y} = \emptyset.$$
Therefore, Assumptions 3.2 (ii) holds. Similarly, since $\mathscr{Y} - \mathrm{Int}\,(\rho) = \emptyset$ then $\partial\mathrm{Im}\,(\rho) = \emptyset$. Since the Cartesian product between any set and an empty-set is the empty-set, then let
$$U \triangleq \emptyset = \partial\mathrm{Im}\,(\rho) = \partial\mathrm{Im}\,(\rho) \times [0,1).$$
Therefore, $\psi(x) = \emptyset$ satisfies Assumption 3.2 (ii). By assumption, $\rho$ is continuous and has a section. Therefore Assumption 3.2 (i.a) holds. Thus, Assumptions 3.2 holds.

Likewise, $\phi$ was taken to be continuous and injective. Therefore, Assumption 3.1 holds. Thus, the result follows from Theorem 3.3 since $\mathscr{F}$ is dense in $C(\mathbb{R}^m, \mathbb{R}^n)$.

Note, that if $\phi$ and $\rho$ are homeomorphisms then both are continuous bijections. Moreover, $\rho$ has a continuous two-sided inverse, and in particular, a continuous section. $\qquad\square$

**Proposition B.5** (Homeomorphic case is Sharp)**.** *Let $\phi$ and $\rho$ be homeomorphisms. Then $\mathscr{F}$ is dense in $C(\mathbb{R}^m, \mathbb{R}^n)$ if and only if $\mathscr{F}_{\rho,\phi}$ is dense in $C(\mathscr{X}, \mathscr{Y})$.*

*Proof.* Since every homeomorphism $\rho$ has a continuous section, namely $\phi^{-1}$, then by Corollary 3.6 if $\mathscr{F}$ is dense in $C(\mathbb{R}^m, \mathbb{R}^n)$ then so is $\mathscr{F}_{\rho,\phi}$. Since $\phi$ is a homeomorphism and $\mathbb{R}^m$ is a locally-compact Hausdorff space then $\mathscr{X}$ is locally-compact and since $\rho$ is also a homeomorphism then similarly $\mathscr{Y}$ is a locally-compact Hausdorff space, since $\mathbb{R}^n$ is. Therefore, by [43, Exercise 46.10]the maps $\Phi_{\rho,\phi} : C(\mathbb{R}^m, \mathbb{R}^n) \to C(\mathscr{X}, \mathscr{Y})$ mapping $f \mapsto \rho \circ f \circ \phi$ and $\Psi_{\rho,\phi} : C(\mathscr{X}, \mathscr{Y})$ mapping $g \mapsto \rho^{-1} \circ g \circ \phi^{-1}$ are continuous with respect to the compact-open topology, because $\rho, \rho^{-1}, \phi,$ and $\phi^{-1}$ are all continuous. Observe that $\Phi_{\rho,\phi} \circ \Psi_{\rho,\phi} = 1_{C(\mathscr{X}, \mathscr{Y})}$, the identity map on $C(\mathscr{X}, \mathscr{Y})$, and $\Psi_{\rho,\phi} \circ \Phi_{\rho,\phi} = 1_{C(\mathbb{R}^m, \mathbb{R}^n)}$, the identity map on $C(\mathbb{R}^m, \mathbb{R}^n)$. Hence, $\Phi_{\rho,\phi}$ is a homeomorphism with inverse $\Psi_{\rho,\phi}$. Now, since $\mathscr{F}_{\rho,\phi} = \Phi(\mathscr{F})$, $\mathscr{F} = \Psi_{\rho,\phi}(\mathscr{F}_{\rho,\phi})$, and since homeomorphisms take dense sets to dense sets then $\mathscr{F}$ is dense in $C(\mathbb{R}^m, \mathbb{R}^n)$ if and only if $\mathscr{F}_{\rho,\phi}$ is dense in $C(\mathscr{X}, \mathscr{Y})$. $\qquad\square$

**Corollary.** *If $\phi$ is a continuous injective map, $\rho$ is a continuous surjection with a section, $\mathscr{X}$ is connected and simply connected, and $\mathscr{F}$ is dense in $C(\mathbb{R}^d, \mathbb{R}^D)$ then $\mathscr{F}_{\rho,\phi}$ is dense in $C(\mathscr{X}, \mathscr{Y})$.*

*Proof.* The proof is identical to the proof of Corollary 3.8 with the only difference that Assumption 3.2 (i.b) holds by assumption instead of Assumption 3.2 (i.a). $\qquad\square$

We now establish Theorem 3.10.

**Theorem** (Geometric Version)**.** *Let $\mathscr{Y}$ be a metrizable manifold with boundary, for which $\mathrm{Int}\,(\mathscr{Y})$ is a smooth manifold, $\mathscr{X}$ is locally-compact, and $\mathscr{F}$ is dense in $C(\mathbb{R}^m, \mathbb{R}^n)$. If $\phi$ satisfies Assumption 3.1 and $\rho$ satisfies Assumption 3.9 then $\mathscr{F}_{\rho,\phi}$ is dense in $C(\mathscr{X}, \mathscr{Y})$.*

*Proof.* For the first portion of the proof we show that $C(\mathbb{R}^n, \mathrm{Im}\,(\rho))$ is dense in $C(\mathbb{R}^n, \mathscr{Y})$. This is achieved in the following steps. First, $C(\mathbb{R}^n, \mathrm{Im}\,(\rho))$ is shown to be dense in $C(\mathbb{R}^n, \mathrm{Int}\,(\mathscr{Y}))$, then that $C(\mathbb{R}^n, \mathrm{Int}\,(\mathscr{Y}))$ is dense in $C(\mathbb{R}^n, \mathscr{Y})$, and then by the transitivity of density it follows that that $C(\mathbb{R}^n, \mathrm{Im}\,(\rho))$ is dense in $C(\mathbb{R}^n, \mathscr{Y})$.

Since $\mathbb{R}^n$ and $\mathrm{Int}\,(\mathscr{Y})$ are smooth manifolds without boundary then, [25, Theorem 2.2] implies that $C^{\infty}(\mathbb{R}^n, \mathscr{Y})$ is dense in $C(\mathbb{R}^n, \mathscr{Y})$ for a strictly stronger topology than the topology of uniform convergence on compacts. Thus, in particular, $C^{\infty}(\mathbb{R}^n, \mathscr{Y})$ is dense in $C(\mathbb{R}^n, \mathscr{Y})$ for the topology of uniform convergence on compacts. By the transitivity of density it is therefore sufficient to show, under Assumption 3.9, that $C^{\infty}(\mathbb{R}^n, \mathrm{Im}\,(\rho))$ is dense in $C^{\infty}(\mathbb{R}^n, \mathrm{Int}\,(\mathscr{Y}))$ to conclude that $C(\mathbb{R}^n, \mathrm{Im}\,(\rho))$ is dense in $C(\mathbb{R}^n, \mathrm{Int}\,(\mathscr{Y}))$.

If $\mathrm{Int}\,(\mathscr{Y}) = \mathrm{Im}\,(\rho)$ then the claim holds vacuously. Therefore, assume that $\mathrm{Int}\,(\mathscr{Y}) \neq \mathrm{Im}\,(\rho)$. By definition, a smooth map $f : \mathbb{R}^n \to \mathrm{Int}\,(\mathscr{Y})$ is transverse to some $\mathrm{Int}\,(\mathscr{Y}) - \mathrm{Im}\,(\rho)$ (i.e.: the inclusion map $\iota_{\mathrm{Int}(\mathscr{Y})-\mathrm{Im}(\rho)} : \mathrm{Int}\,(\mathscr{Y}) - \mathrm{Im}\,(\rho) \to \mathrm{Int}\,(\mathscr{Y}))$ if for every $y \in \mathrm{Int}\,(\mathscr{Y}) - \mathrm{Im}\,(\rho)$
$$\mathrm{Im}\,(df_x) + T_{f(x)}(\mathrm{Int}\,(\mathscr{Y}) - \mathrm{Im}\,(\rho)) = T_{f(x)}(\mathrm{Int}\,(\mathscr{Y})). \tag{21}$$

However, Assumption 3.9 (ii) requires that $\dim(\text{Int}(\mathscr{Y})) - \dim(\text{Int}(\mathscr{Y}) - \text{Im}(\rho)) > n$, which implies that (21) can only hold when, for each $y \in \text{Int}(\mathscr{Y}) - \text{Im}(\rho)$

$$\{0\} = \text{Im}(df_x) \cap T_{f(x)}(\text{Int}(\mathscr{Y}) - \text{Im}(\rho)). \tag{22}$$

By [22, Chapter 1, Excersize 4] , (22) implies that $f(\mathbb{R}^n) \cap \text{Int}(\mathscr{Y}) - \text{Im}(\rho) = \emptyset$. Therefore, $f \in C^\infty(\mathbb{R}^n, \text{Int}(\mathscr{Y}))$ is transversal to $\text{Int}(\mathscr{Y}) - \text{Im}(\rho)$ only if $f \in C(\mathbb{R}^n, \text{Im}(\rho))$. Since the set of all $f \in C^\infty(\mathbb{R}^n, \text{Int}(\mathscr{Y}))$ is dense in $C^\infty(\mathbb{R}^n, \text{Int}(\mathscr{Y}))$, by [25, Theorem 2.1], then $C^\infty(\mathbb{R}^n, \text{Im}(\rho))$ is dense in $C^\infty(\mathbb{R}^n, \text{Int}(\mathscr{Y}))$. Consequentially, $C(\mathbb{R}^n, \text{Im}(\rho))$ is dense in $C(\mathbb{R}^n, \text{Int}(\mathscr{Y}))$.

Consider the inclusion map $\iota_{\text{Int}(\mathscr{Y})} \to \mathscr{Y}$. Then $\iota_{\text{Int}(\mathscr{Y})}$ is an embedding of $\text{Int}(\mathscr{Y})$ into with section the identity map and it is dense in $\mathscr{Y}$. Therefore, Assumption 3.2 (i)-(ii) hold. Assumption 3.2 (iii) is precisely the definition of a $\partial \text{Im}(\iota_{\text{Int}(\mathscr{Y})}) = \partial \text{Int}(\mathscr{Y})$ being collared (see [8, Section II; page 332]). By definition, $\partial \text{Int}(\mathscr{Y})$ is the *boundary*, in the sense of manifolds with boundary, of the manifold with boundary $\mathscr{Y}$. Since $\iota_{\text{Int}(\mathscr{Y})}$ is a surjection onto $\text{Int}(\mathscr{Y})$ then Assumption 3.2 (iii) states that the boundary of the manifold with boundary $\mathscr{Y}$ must be collared. However, since $\mathscr{Y}$ is metrizable then this is guaranteed by [8, Theorem 2]. Thus, Assumption 3.2 (iii) holds. Therefore, Assumption 3.2 holds and Lemma B.4 guarantees that $C(\mathbb{R}^n, \text{Int}(\mathscr{Y}))$ is dense in $C(\mathbb{R}^n, \mathscr{Y})$. Hence, $C(\mathbb{R}^n, \text{Im}(\rho))$ is dense in $C(\mathbb{R}^n, \mathscr{Y})$.

Since $\mathscr{F}$ is dense in $C(\mathbb{R}^m, \mathbb{R}^n)$, since the identity map satisfies Assumption 3.1, then Assumption 3.2 implies that Lemma B.3 applies. Whence, $\mathscr{F}_{\rho, 1_{\mathbb{R}^m}}$ is dense in $C(\mathbb{R}^n, \text{Im}(\rho))$. Therefore, $\mathscr{F}_{\rho, 1_{\mathbb{R}^m}}$ is dense in $C(\mathbb{R}^n, \mathscr{Y})$. Applying Lemma B.2, we conclude that $\mathscr{F}_{\rho, \phi}$ is dense in $C(\mathscr{X}, \text{Im}(\rho))$ and therefore in $C(\mathscr{X}, \mathscr{Y})$ for the compact-open topology by Lemma B.4.

Finally, notice that, since Assumptions 3.2 (i)-(ii) were assumed to hold then Theorem 3.3 implies that $\mathscr{F}_{\rho, \phi}$ is dense in $C(\mathscr{X}, \mathscr{Y})$ for the co topology. Since every singleton is closed in $\mathbb{R}^m$ and $\phi$ is continuous and injective then every $x \in \mathscr{X}$, $\{x\}$ is the continuous pre-image of the singleton $\{\phi(x)\} \in \mathbb{R}^m$ by $\phi$. Since $\phi$ is continuous, then $\{x\}$ is closed therefore [43, Theorem 17.8] implies that $\mathscr{X}$ is Hausdorff. Since $\mathscr{X}$ was assumed to be locally-compact and $\mathscr{Y}$ is metrizable then [43, Theorem 46.8] implies that the co topology and the ucc topology on $C(\mathscr{X}, \mathscr{Y})$ coincide for any metric topologizing $\mathscr{Y}$. □

**Theorem B.6** (Universal Classification: General Case). *Let $\{0,1\}^n$ be equipped with the n-fold product of the Sierpiński topology on $\{0,1\}$, $\phi$ satisfy Assumption 3.1, $\rho : \mathbb{R}^n \to (0,1)^n$ be a homeomorphism, $\alpha \in (0,1)$, and $\mathscr{F} \subseteq C(\mathbb{R}^m, \mathbb{R}^n)$ be dense. Let $\{\mathscr{X}_i\}_{i=1}^n$ be a set of open subsets of a metric space $\mathscr{X}$ and let $\hat{h}$ be its associated ideal classifier defined by* (6). *Then the following hold:*

*(i) (Hard-Soft Decomposition) There exist continuous functions $\hat{s}_i \in C(\mathscr{X}, [0,1])$ such that*

$$\hat{h} = I_{(\alpha,1]} \bullet (s_1, \ldots, s_n) \qquad \hat{s}_i^{-1}[(\alpha,1]] = \mathscr{X}_i, (\forall i = 1, \ldots, n)$$

*(ii) (Universal Classification) There exists a sequence $\{f_k\}_{k \in \mathbb{N}}$ in $\mathscr{F}$ such that:*

*(a) (Soft Classification) For each non-empty compact subset $\kappa \subseteq \mathscr{X}$ and every $\varepsilon > 0$, there is some $K \in \mathbb{N}^+$ such that*

$$\sup_{x \in \kappa} \max_{i=1,\ldots,n} |\rho \circ f_k \circ \phi(x)_i - \hat{s}_i(x_i)| < \varepsilon, \qquad (\forall k \geq K)$$

*(b) (Hard Classification) $I_{(\alpha,1]} \bullet \rho \circ f_k \circ \phi$ converges to $\hat{h}$ in $C(\mathscr{X}, \{0,1\}^n)$ for the co-topology.*

*Furthermore, $\mathscr{F}_{\rho, \phi}$ is dense in $C(\mathscr{X}, [0,1]^n)$.*

*Proof.* Since $\mathscr{X}$ is a metric space and since each $\mathscr{X}_i$ is open then by [2, Corollary 3.19] $\mathscr{X}_i$ is an open $F_\sigma$ set, i.e.: an open set which is the countable intersection of closed sets. By [15, Corollary 1.5.13] there exists continuous function $\tilde{s}_i : \mathscr{X} \to [0,1]$ such that $\tilde{s}_i^{-1}[(0,1]] = \mathscr{X}_i$. Since $\alpha \in (0,1)$, then for each $i = 1, \ldots, n$, let $s_i(x) = \alpha + (1-\alpha)\tilde{s}_i(x)$. Then, $s_i$ is continuous from $\mathscr{X}$ to $[0,1]$ and satisfies $s_i^{-1}[(\alpha,1]] = \mathscr{X}_i$. Define $\hat{s} \triangleq (s_1, \ldots, s_n)$. Note $\hat{s}$ is continuous from $\mathscr{X}$ to $[0,1]^n$ since its components continuously map $\mathscr{X}$ to $[0,1]$. Next, define the map

$$\Phi_\alpha : [0,1]^n \to \{0,1\}^n$$
$$x \to (I_{(\alpha,1]}(x_i))_{i=1}^n,$$

and note that $\Phi \circ \hat{s} = I_{(\alpha,1]} \bullet \hat{s} = \hat{h}$, by construction. Thus, (i) holds.

Since $\phi$ is an embedding of $\mathscr{X}$ into $\mathbb{R}^m$ for which $\phi(\mathscr{X})$ is a retract, $\rho$ is a homeomorphism of $\mathbb{R}^n$ onto $(0,1)^n$, and since $[0,1]^n$ is a metrizable manifold with boundary whose interior is $\mathrm{Int}\left([0,1]^n\right) = (0,1)^n$ then Theorem 3.10 implies that $\mathscr{F}_{\rho,\phi}$ is dense in $C(\mathscr{X},[0,1]^n)$.

In particular, this implies that, since $\hat{s} \in C(\mathscr{X},[0,1]^n)$, and $\mathscr{F}_{\rho,\phi}$ is dense therein, then there exists a sequence $\{f_k\}_{k\in\mathbb{N}}$ in $\mathscr{F}$ such that, for every non-empty compact-subset $\kappa \subseteq \mathscr{X}$, every $\tilde{\varepsilon} > 0$

$$\sup_{x\in\kappa} \|\rho \circ f_k \circ \phi(x)_i - s_i(x_i)\| < \tilde{\varepsilon} \qquad (\forall k \geq K). \tag{23}$$

Applying [12, Theorem 3.1] to (23) yields a constant $C > 0$, independent of $x, \{f_k\}_{k\in\mathbb{N}}$, and of $\hat{s}$, satisfying

$$\sup_{x\in\kappa} \max_{i=1,\dots,n} |\rho \circ f_k \circ \phi(x)_i - s_i(x_i)| \leq C \sup_{x\in\kappa} \|\rho \circ f_k \circ \phi(x)_i - s_i(x_i)\| < C\varepsilon \qquad (\forall k \geq K).$$

Setting $\varepsilon \triangleq C\tilde{\varepsilon}$ yields (ii.a). Thus, we only need to verify (ii.b).

Equip $\{0,1\}$ with the Sierpiński topology $\{\emptyset, \{1\}, \{0,1\}\}$ and denote this space by $S$. Then, for any $T_1$ space, and any open set $U$, the indicator function $I_U$ of $U$ is continuous to $\{0,1\}$, see [54, Chapter 7]. In particular, since $[0,1]$ with the Euclidean topology is a metric space, then in particular, it is $T_1$. Moreover, for any $\alpha \in (0,1)$, the set $(\alpha,1]$ is open in $[0,1]$. Therefore, the map $I_{(\alpha,1]} : [0,1] \to S$ is continuous and thus the map $\Phi_\alpha$ is continuous. By [43], since post-composition by continuous functions defines a continuous map between $C(\mathscr{X},[0,1]^n)$ and $C(\mathscr{X},\{0,1\}^n)$ when both are equipped with the compact-open topology then the map

$$\Phi : C(\mathscr{X},[0,1]^n) \to C(\mathscr{X},\{0,1\}^n) \atop f \to \Phi_\alpha \circ f,$$

is continuous. Since continuous functions preserve convergent sequences and since $\{\rho \circ f_k \circ \phi\}_{k\in\mathbb{N}}$ converges to $\hat{s}$ in the compact-open topology on $C(\mathscr{X},[0,1]^n)$ then

$$\{\Phi(\rho \circ f_k \circ \phi)\}_{k\in\mathbb{N}} = \{\Phi_\alpha \circ \rho \circ f_k \circ \phi\}_{k\in\mathbb{N}} = \{I_{(\alpha,1]} \bullet \rho \circ f_k \circ \phi\}_{k\in\mathbb{N}}$$

converges to $\Phi(\hat{s}) = \Phi_\alpha \circ \hat{s} = I_{(\alpha,1]} \bullet \hat{s}$ in the compact-open topology on $C(\mathscr{X},0,1^n)$. This verifies (ii.b). Thus, (ii) holds. $\qquad\square$

**Corollary** (Universal Classification: Deep Feed-Forward Networks). *Let $\{\mathscr{X}_i\}_{i=1}^n$ be open subsets of $\mathscr{X}$, and $\hat{h}$ be their associated ideal classifier. Let $\phi : \mathscr{X} \to \mathbb{R}^n$ be a continuous injective feature map. Let $\sigma$ be a continuous, locally-bounded, and non-constant activation function. Let $\rho$ either be the component-wise logistic function. Then there exists a sequence $\{f_k\}_{k\in\mathbb{N}^+}$ of DNNs satisfying the conclusions of Theorem 3.11.*

*Proof.* By [36] the set $\mathscr{N}\mathscr{N}^\sigma$ of all feed-forward DNNs is dense in $C(\mathbb{R}^m, \mathbb{R}^n)$. Moreover, the identity map $1_{\mathbb{R}^m}$ on $\mathbb{R}^m$ satisfies Assumption 3.1. Furthermore, both the soft-max

$$x \to \left(\frac{e^{x_j}}{\sum_{i=1}^n e^{x_i}}\right)_{j=1}^n$$

and the component-wise logistic function

$$x \to \left(\frac{e^{x_j}}{1+e^{x_j}}\right)_{j=1}^n,$$

are continuous bijections with continuous inverses, from $\mathbb{R}^n$ onto $(0,1)^n$. In particular, they satisfy Assumption 3.2. Therefore Theorem 3.11 applies and the conclusion holds. $\qquad\square$

**Corollary** (Universal Classification: Deep CNNs). *Let $2 \leq s \leq n$, $\{\mathscr{X}_i\}_{i=1}^n$ be open subsets of $\mathscr{X}$, and $\hat{h}$ be their associated ideal classifier. Let $\phi : \mathscr{X} \to \mathbb{R}^n$ be a continuous injective feature map and let $\rho : \mathbb{R} \to (0,1)$ be the logistic function. Then there is a sequence of deep CNNS $\{f_k\}_{k\in\mathbb{N}^+}$ in $Conv_{\rho,\phi}^s$ satisfying the conclusion of Theorem 3.11.*

*Proof.* Since $Conv^s$ with these specifications is dense in $C(\mathbb{R}^d, \mathbb{R})$ by [57], $\phi$ is a continuous injection, and $\rho$ is a homeomorphism of $\mathbb{R}$ onto $(0,1)$ then the result follos from Theorem 3.11. $\qquad\square$

**Corollary** (Cartan-Hadamard Version). *Let $\mathscr{F}$ be dense in $C(\mathbb{R}^m, \mathbb{R}^n)$, let $\mathscr{M}$ and $\mathscr{N}$ be Cartan-Hadamard manifolds of dimension $m$ and $n$. Then, $\mathscr{F}_{\mathrm{Log}_p^{\mathscr{M}}, \mathrm{Exp}_q^{\mathscr{N}}}$ is dense in $C(\mathscr{M}, \mathscr{N})$.*

*Proof.* Since $\mathscr{N}$ and $\mathscr{M}$ are Hadamard manifolds then the Cartan-Hadamard Theorem, see [31, Corollary 6.9.1], implies that $\mathrm{Exp}_p^{\mathscr{M}}$ and $\mathrm{Log}_q^{\mathscr{N}}$ are diffeomorphisms and in particular are homeomorphisms. Since $\mathscr{N}$ and $\mathscr{M}$ have no boundary then the result follows from Corollary 3.8. $\qquad\square$

**Corollary** (Universal Approximation for Symmetric Positive-Definite Matrices). *Let $d, D \in \mathbb{N}^+$ and $\mathscr{F} \subseteq C(\mathbb{R}^{d(d+1)/2}, \mathbb{R}^{D(D+1)/2})$ be ucc-dense. Then, for any $A \in P_d^+$ and $B \in P_D^+$, $\mathscr{F}_{\mathrm{Log}_A, \mathrm{Exp}_B}$ is ucc-dense in $C(P_d^+, P_D^+)$. In particular, if $\sigma$ is a continuous, locally-bounded, and non-polynomial activation function then $\mathscr{NN}^{\sigma}_{\mathrm{Log}_A, \mathrm{Exp}_B}$ is ucc-dense in $C(P_d^+, P_D^+)$.*

*Proof.* As discussed in [7, Section 3.3], the space $P_d^+$ under the metric $d_+$ is a complete, connected, and simply connected Riemannian manifold with non-positive curvature. Therefore, Corollary 3.14 applies. $\qquad\square$

**Corollary** (Hyperbolic Neural Networks are Universal). *Let $\sigma$ be a continuous, non-polynomial, locally-bounded activation function and $c > 0$. Then for every $g \in C(\mathbb{D}_c^m, \mathbb{D}_c^n)$, every $\varepsilon > 0$, and every compact subset $K \subseteq \mathbb{D}_c^m$ there exists a hyperbolic neural network $g_{\varepsilon, K, c}$ in (10) satisfying*

$$\sup_{x \in K} d_c(g(x), g_{\varepsilon, K, c}) < \varepsilon.$$

*Proof.* Since $\mathbb{D}_c^n$ is a complete, connected, simply connected, of non-positive sectional curvature Riemannian manifold then it is of Cartan-Hadamard type. Therefore, the result follows directly form Corollary 3.14. $\qquad\square$

**Theorem.** *If Assumption 3.17 holds, then there exists a measurable subset $\Omega' \subseteq \left\{ \omega \in \Omega : \overline{\mathscr{NN}^{\sigma}_{2,k}(\omega)} = C(\mathbb{R}^d, \mathbb{R}^D) \right\}$ satisfying $\mathbb{P}(\Omega') = 1$.*

*Proof.* Let $\mathrm{Mat}_{k,l}$ denote the collection of $k \times l$ matrices with real coefficients, identified with the Euclidean space $\mathbb{R}^{kl}$. By Assumption 3.17 (ii) $\sigma$ is a injective continuous function. Therefore so is the function

$$\sigma_i : \mathbb{R}^{d_{i+1}} \to \mathbb{R}^{d_{i+1}}$$
$$(x_j)_{j=1}^{d_{i+1}} \mapsto (\sigma(x_j))_{j=1}^{d_{i+1}}.$$

Fix $\omega \in \Omega$. Since $d_i = d_{i+1}$, then the map $W_i(\cdot, \omega)$ is affine injection if and only if $A_i(\omega)$ is of rank $d_i$. Since the composition of continuous injections is again continuous then the map

$$\phi(x, \omega) : \mathbb{R}^d \to \mathbb{R}^{d_k} \tag{24}$$
$$x \mapsto \Sigma_k \circ A_k(x, \omega) \circ \cdots \circ \Sigma_1 \circ A_1(x, \omega),$$

is continuous and injective if each $A_i(\omega)$ is of full rank. Equivalently, it is enough to show that the smallest singular value of $A_i$, which we denote by $\lambda^{\star}(A_i)$ to avoid confusion with the notation for activation functions, should be bounded away from 0 with probability 1.

Since each $A_i$ is a random $d_{i+1} \times d_i$ matrix and $d_{i+1} \geq d_i$, then it contains a random $d_i \times d_i$ (square) sub-matrix. It is enough to show that this random sub-matrix is of full-rank. Therefore, without loss of generality, we assume that $d_{i+1} = d_i$ for all $i = 1, \ldots, k$.

Since each $A_i$ is a random $d_i \times d_i$ square matrix and Assumptions 3.17 hold then [53, Theorem 1.3 (Universality for the Least Singular Value)] applies whence. Therefore,

$$
\begin{aligned}
\mathbb{P}\left(\lambda^{\star}(A_i) > 0 \,\forall (i = 1, \ldots, k)\right) &= 1 - \prod_{i=1}^{k} \mathbb{P}\left(\lambda^{\star}(A_i) > 0\right) \\
&= 1 - \lim_{t \downarrow 0^+} \prod_{i=1}^{k} \mathbb{P}\left(\lambda^{\star}(A_i) \leq \frac{t}{\sqrt{d_i}}\right) \\
&= \left( \int_0^t \frac{1 + \sqrt{x}}{2\sqrt{x}} e^{-\frac{x}{2} - \sqrt{x}} dx + O\left(\frac{1}{x^c}\right) \right)^k \\
&= 0,
\end{aligned}
$$

where $c > 0$ is an absolute constant independent of $\xi$. Therefore, the set

$$\Omega' \triangleq \{\omega \in \Omega : \lambda_i(A_i) > 0 \,\forall i = 1, \dots, k\} \subseteq \left\{\omega \in \Omega : \overline{\mathcal{N}\mathcal{N}^\sigma(\omega)} = C(\mathbb{R}^d, \mathbb{R}^D)\right\}$$

is $\Sigma$-measurable and $\mathbb{P}(\Omega') = 1$. This concludes the proof. $\qquad\square$

**Corollary** (Sub-Gaussian Case with Sigmoid Activation). *Let $X_i = Z_i$ for each $i = 1, \dots, k$ be independent sub-Gaussian random-variables, $\sigma(x) = \frac{1}{1+e^{-x}}$, and $d_i = d$ for each $i = 1, \dots, k$. Then the conclusion of Theorem 3.18 holds.*

*Proof.* Since the $X_i$ and $Z_i$ are sub-Gaussian random variables, then by [9] all their moments are finite. This verifies Assumption 3.17 (iv). Since they are assumed to be standardized then Assumption 3.17 (iii) holds. Since the sigmoid activation function is continuous and monotonically increasing then Assumption 3.17 (ii) holds. Lastly, Assumption 3.17 (i) holds by construction. Therefore, Theorem 3.18 applies and the conclusion follows. $\qquad\square$

**Corollary** (Bernoulli Case with PReLU Activation). *Suppose that for every $i, j = 1, \dots, k$, $X_i$ and $Z_j$ i.i.d. copies of a random variable taking values $\{-1, 1\}$ with probabilities $\{\frac{1}{2}, \frac{1}{2}\}$. Let $d_i = d$ for each $i = 1, \dots, k$ and $\sigma$ be the PReLU activation function of [23]. Then Assumptions 3.17 are met; thus the conclusion of Theorem 3.18 holds.*

*Proof.* Since the PReLU activation function is continuous and strictly increasing, then Assumption 3.17 (ii) holds. Since, Bernoulli random variables have all finite $C^{th}$-moments, for $C > 0$, then Assumption 3.17 (iv) holds. Since $X_i$ and $Z_i$ are taken to be standardized, then Assumption 3.17 (iii) holds. Assumption 3.17 (i) holds by hypothesis. Therefore, the result follows from Theorem 3.18. $\qquad\square$

**Proposition.** *Let $\sigma$ be a continuous and strictly increasing activation function, $J \in \mathbb{N}_+$, $A_1, \dots, A_J$ be $m \times m$ matrices, and $b_1, \dots, b_J \in \mathbb{R}^d$. Let $\phi(x) \triangleq \phi_J(x)$ where*

$$\phi_j(x) \triangleq \sigma \bullet \left(\exp(A_j)\phi_{j-1}(x) + b_j\right) \qquad \phi_0(x) \triangleq x, \quad j = 1, \dots, J, \tag{25}$$

*where* exp *is the matrix exponential. Then $\phi$ satisfies Assumption 3.1.*

*Proof.* Since $\sigma$ is strictly increasing then it is injective. Therefore, the map $\phi_1(x) \triangleq (\sigma(x_k))_{k=1}^m$ is continuous and injective from $\mathbb{R}^m$ to itself. Next, for each $1 \leq j \leq J$ with $j \in \mathbb{N}_+$, each $A_j$ is a $d \times d$ matrix then its exponential $\exp(A_j)$ is in the general linear group (see [33]) and therefore it is an invertible $m \times m$ matrix; hence, the map $\phi_{2,j}(x) \triangleq \exp(A_j)x$ is a continuous bijection from $\mathbb{R}^m$ to itself. Finally, for each $b_j \in \mathbb{R}^m$, the map $\phi_3(x) \triangleq x + b_j$ is a continuous bijection, since it is affine with inverse $y \mapsto y - b_j$; hence, $\phi_{3,j}$ is a continuous injection from $\mathbb{R}^m$ to itself. Since the composition of continuous functions is again continuous and the composition of injective functions is again injective then the map $\phi = \phi_1 \circ \phi_{3,J} \circ \phi_{2,J} \circ \cdots \circ \phi_1 \circ \phi_{3,1} \circ \phi_{2,1}$ of (25) is a continuous injection; hence, it satisfies Assumption 3.1. $\qquad\square$

**Proposition.** *In the setting of Proposition 3.21, if $\sigma$ is also surjective then $\phi$ is a homeomorphism, and in particular it satisfies Assumption 3.2.*

*Proof.* By Proposition 3.21 the map $\phi$ is continuous and injective. Since $\sigma$ is a continuous strictly increasing function then it is injective. Moreover by [26] it is a homeomorphism onto its image. Since $\sigma$ was assumed to be a surjection then this means that $\sigma$ is a homeomorphism from $\mathbb{R}$ to itself. Therefore, the map $\prod_{i=1}^m \sigma : \mathbb{R}^m \to \mathbb{R}^m$ sending $x$ to $(\sigma(x_i))_{i=1}^m$ is a homeomorphism by [43, Theorem 19.6]. In the proof of Proposition 3.21, above, it was shown that for each $1 \leq j \leq J$, $j \in \mathbb{N}_+$, the maps $x \mapsto \exp(A_j)x$ and $x \mapsto x + b_j$ are homeomorphisms. Therefore, since the composition of homeomorphisms is again a homeomorphism then $\phi$ is a homeomorphism. In particular, it satisfies Assumption 3.2 since $\phi^{-1}$ exists and is continuous (thus Assumption 3.2 (i) holds), $\operatorname{Im}\phi = \mathbb{R}^m$ (thus Assumption 3.2 (ii) holds), and since $\partial \operatorname{Im}\phi = \emptyset$ (thus Assumption 3.2 (iii) holds). $\qquad\square$

**Proposition** (Graphs of Continuous Functions are Good Feature Maps). *Let $d \in \mathbb{N}_+$ and $g \in C(\mathbb{R}^m, \mathbb{R}^d)$. Then $\phi_g(x) \triangleq (x, g(x))$ satisfies Assumption 3.1.*

*Proof.* Let identity map $1_{\mathbb{R}^m}(x) \triangleq x$ is continuous from $\mathbb{R}^m$ to itself and by hypothesis the map $g : \mathbb{R}^m \to \mathbb{R}^d$ is continuous, therefore, by [43, Theorem 19.6] the map $1_{\mathbb{R}^m} \times g : \mathbb{R}^m \times \mathbb{R}^m \to \mathbb{R}^m \times \mathbb{R}^d = \mathbb{R}^{m+d}$ is continuous. A consequence of the same result shows that, for topological spaces $X, Y$, and $Z$ and a function $F : Z \to X \times Y$ is continuous, when $X \times Y$ is equipped with the product topology, if and only if both $\pi_1 \circ F : Z \to X$ and $\pi_2 \circ F : Z \to Y$ are continuous, where $\pi_1(x, y) = x$ and $\pi_2(x, y) = y$. Therefore, when $X = Y = Z = \mathbb{R}^m$ and $F(x) \triangleq (x, x)$ then

$$\pi_1 \circ F = 1_{\mathbb{R}^m} = \pi_2 \circ F, \tag{26}$$

and since $1_{\mathbb{R}^m}$ is continuous then (26) implies that $F$ is continuous. Since the composition of continuous functions is again continuous, then $(1_{\mathbb{R}^m} \times g) \circ F(x) = (x, g(x))$ is continuous from $\mathbb{R}^m$ to $\mathbb{R}^{m+d}$. Lastly, notice that if $x_1 \neq x_2$ in $\mathbb{R}^m$ then $\|x_1 - x_2\| > 0$ and therefore

$$\|(x_1, g(x_1)) - (x_2, g(x_2))\| \geq \|x_1 - x_2\| > 0.$$

Hence, $(x_1, g(x_1)) \neq (x_2, g(x_2))$ in $\mathbb{R}^{m+d}$ and therefore, $(1_{\mathbb{R}^m} \times g) \circ F$ is a continuous and injective functions. Therefore, Assumption 3.1 holds. □