[Reviews · NeurIPS 2020]

Review 1

Summary and Contributions: This paper discusses "which modifications to the input and output layers of a neural network architecture preserve its universal approximation capabilities" and proposes some interesting theories and lemma.

Strengths: They proposed a non-Euclidean universal approximation capabilities. Furthermore, it generalizes the discussion to different machine frameworks. Some conclusion may be useful for guiding people who want to seek some theoretical support about universal approximation.

Weaknesses: It is far away from practical implementation. The angle which the paper starts from is to analysis non-Euclidean modifications while almost all the given results are what conditions the modifications should meet. However, it lack of necessary experiments to support the author's conclusion and viewpoints. After all, the mathematical conclusion holds when some minor terms are neglected from the forumula. Whereas in real application, especially in the era of deep learning, these minor terms may play a crucial role in improving predction performance . Therefore, the authors should provide some simulated and practical experiments to support their conclusion. Minor Issue: Not enough emphasis in introduction. Embolden the important conclusions is a good choice such as the application in line 56 to 57.

Correctness: I am not sure whether the proposped methodology is correct because the underlying mathematical theory is far away from my major. I can only provide some feeling of mine from machine learning but not math viewpoint.

Clarity: If there are several experiments on real-world datasets, maybe it is better for helping readers understand the contribution of this paper. A pure theoretical conclusion does not bring important and sufficient guidance to the readers.

Relation to Prior Work: should be

Reproducibility: No

Additional Feedback: no experiments. Unlike other papers, this paper's style does not match that of machine learning. I am ok if this paper is accepted.


Review 2

Summary and Contributions: This is a learning theory paper that studies under what conditions, modifying input and output space will keep the universal approximation of neural networks. In particular, this paper argues that if the feature map regularity (Assumption 3.1) and readout map regularity (Assumption 3.2) are both satisfied, then the universal approximator is dense in the compact sets. In addition, this paper provides implications in geometric deep learning, which often deals with non-Euclidean data. So this paper will potentially provide insights for many applications, such as health care, social networks, and natural science. Also, the problem studied in this paper is novel and might encourage many empirical works to study the same problem. ======================================================= I thank authors to provide the rebuttal. My review doesn't change after considering the rebuttal and other reviews.

Strengths: 1. This paper studies an important problem -- how to preserve the universal approximation of machine learning models while changing their input and output spaces. This is a useful problem when transferring architecture between different machine learning tasks is desired. 2. Theoretically, this paper provides sufficient conditions for the input/output spaces as well as the feature maps. 3. This paper provides its implications for many related topics, such as random features and geometric deep learning.

Weaknesses: 1. I understand this paper is a learning theory paper. However, I still think having some experiments (even they are small empirical studies) should make this paper stronger and empirically convincing. 2. Empirically, modifying neural network architecture for a certain task is a very common practice, which also involves changing internal structure of the architecture beyond input and output maps. However, in this paper, it assumes the feature mapping is regular (Assumption 3.1) and doesn't study much of it. So I am worried that the paper is too restrict to input and output mapping. 3. Following 2, when designing a neural network architecture for a specific task, people often inject a particular "inductive bias" to it. So I am wondering, does the universal approximation claim on feature map suffice?

Correctness: The claims and assumptions in this paper seem self-contained to me. This paper doesn't provide empirical method.

Clarity: Overall, this paper is well written. It provides backgrounds for audiences who are not familiar with this field.

Relation to Prior Work: I am not very familiar with prior works. From my understanding, this paper does differ a lot from previous papers. This paper studies a new problem and proposes theorems/corollaries that are not present in the past works.

Reproducibility: Yes

Additional Feedback:


Review 3

Summary and Contributions: This paper is about theoretically understanding the universal approximation capability of neural networks. In detail, the author presents a sufficient and general condition on the input and readout feature maps which is able to preserve the universal approximation capabilities that the internal neural network architecture has. It follows that such condition can be applied to Deep Feed-Forward Networks and Deep CNNs, which shows they are indeed universal classifiers based the conditions proposed. Moreover, The result is applied to the field of geometric deep learning, which proves if the input and output features lie on some manifold (Cartan-Hadamard) with specific conditions, then there exists some input(resp. readout) mapping from the manifold to Euclidean space(resp. from the Euclidean space back to the manifold) which preserves the universal approximation capability of the architecture used. In addition, the author provides a theoretical justification that the Deep networks with randomized layers preserve the universal approximation property with probability one.

Strengths: In general, the result proved in this work appears to be correct. The conclusion on Deep Feed-Forward networks, Deep CNNs as well as the hyperbolic neural networks seems to be interesting to me. Such result might be useful to the future understanding of the universal approximation capability of Neural Networks. Also, some techniques of designing the architecture or training the neural networks could benefits from the result. In detail, the condition (Assumption 3.1 & 3.2) is simple and the general version of the condition seems to be sound. The application on showing that Deep Feed-Forward Networks and Deep CNNs can be universal classifiers under the assumption of the input mapping, appears to be novel. In addition, the extension of such conditions to the manifold is interesting.

Weaknesses: My first concern is that there is no experimental justification of the result proved in this paper, so I think it might be helpful to show some simple experiments on verifying the proved universal approximation capability on deep ffNNs, deep CNNs or hyperbolic neural networks, especially check with the proposed conditions exists or not. This could further indicates the importance of the assumptions and conditions this work proposed. Secondly, there seems to be no much discussion about how the proved results can impact the future development of the field, so I expect there could be some discussion showing the importance of the presented results (this problem can also be alleviated by some experiments as mentioned above). Also, since most of the results presented are based on some assumptions, I think it might be helpful to illustrate more on such assumption by examples, e.g. in some real tasks when does such condition could fail? Adding this could greatly enhance the soundness of the theoretical result proved. It is mentioned in this work that the proposed conditions or assumptions can be easily verified, so I was wondering is there any guideline for building such input and readout mapping satisfying the conditions? How does the proposed conditions on manifold relate to some standard manifold laerning techniques (e.g. Diffusion maps and Laplacian Eigenmaps)? This is just my own question, thanks. There are some statement in this paper seems vague to me (It might be the case that I missed some important points). Line 10: It says the result is obtained on universal Guassian process, while in the main paper there appears to be no result directly point out this, could the author please further explains this? Line 185: Is that open subspace $U_y$? Line 270, Is $X$ should be $\mathcal{X}$?

Correctness: Due to the time limits I may not be able to check all the proofs in the supplementary material. Most arguments in the main paper appear to be correct.

Clarity: Although this paper is overall readable, I believe there is certain space for improvement. My personal suggestions are follwoing: (1) It might be helpful to define some commonly used notations before the main result. E.g., the component-wise composition on line 236 is defined later on line 242. (2) On page 2, I think the discussion on variant implications can be simplified, and the detailed discussion can be deferred to the main result, when introducing the result on each implication separately. BTW, is the `Implications for Classification' on line 84 a paragraph or a subsection? (3) Since this work introduces many definitions and techniques about topology and manifold, which could be difficult to follow easily. I think it might be helpful to illustrate some argument by figures or more examples. E.g., illustrate what is the $Exp_p^M$ and $Log_p^M$ given on line 276 and 277.

Relation to Prior Work: There is no individual related work section in this paper, some related works are discussed in the introduction section. I believe some clarification is needed to make it clear.

Reproducibility: Yes

Additional Feedback: The feedback is appreciated, I think it is helpful to strength this paper by providing more experiments like the one shown in the feedback.


Review 4

Summary and Contributions: The authors deal with lower and upper layers of DNN and are interested in universal approximation properties. More specifically, they focus on classification 'readout' layer and randomized features. The authors produces theoretical results to ensure universal approximation for hyperbolic neural networks. I have read the authors' feedback.

Strengths: Good literature review, related works, etc. Relevant questions are addressed, although some aspects belong to a very specific niche..

Weaknesses: The paper has no clear red line. The title is misleading and should better reflect the paper content. The style and structure are difficult to follow. Somehow one has the feeling that the authors are interested in a very particular type of network architecture, e.g. to deal with covariance matrix and classify them, but they try to get more general results, which is good, and eventually they end up half way. Another style flaw is the some parts of the text read very trivial, like the fact that if a DNN includes a couple of hidden layers, it has UA. Maybe this is not trivial but the intuition that it is true is natural and the authors should better highlight what are the stakes here. Same for the classification readout layer. One can hope that something simple as the softmax has been designed so as not to spoil UA of the previous layers. And similar feeling of triviality for random features, with e.g. ELMs and other radom methods that make the authors' result quite expected and not surprising. There are neither experimental results nor any illustrative example. The broader impact section is far fetched.

Correctness: The mathematical development seem to be correct.

Clarity: See weaknesses. English is good but clarity and structure are issues.

Relation to Prior Work: Excellent.

Reproducibility: No

Additional Feedback: About reproducibility: there are no results to be reproduced.

[Author Response · NeurIPS 2020]

We would like to thank the reviewers for taking the time to carefully read, evaluate, and give feedback on our submission.

**Assumptions 3.1 and 3.2- Reviewer 2:** *"[T]his paper... assumes that the feature map is regular (Assumption 3.1)... I am worried that the [assumptions] are too restrictive".* (Assumption 3.1) is necessary and sufficient for $\{f \circ \phi : f \in \mathcal{F}\}$ to be dense in $C(X, \mathbb{R}^n)$ if $\mathcal{F}$ is. Indeed, if $\phi$ were not continuous then $\{f \circ \phi : f \in \mathcal{F}\}$ may fail to belong to $C(X, \mathbb{R}^n)$. If $\phi$ were not injective then there would exist $x_1, x_2 \in X$ such that $\phi(x_1) = \phi(x_2)$ and therefore any $g \in \{f \circ \phi : f \in \mathcal{F}\}$ satisfies $g(x_1) = g(x_2)$, and likewise for limits of any sequence in $\{f \circ \phi : f \in \mathcal{F}\}$. Hence $\{f \circ \phi : f \in \mathcal{F}\}$ would be a proper closed subset of $C(X, \mathbb{R}^n)$ and therefore it could not be dense. (Assumption 3.2) is almost sharp and a characterization can be obtained using the $\mathcal{Z}$-sets as defined in [1]. However, it is unlikely that a non-pathological example can be generated which fails our assumptions but meets a refinement using $\mathcal{Z}$-sets.

**Examples - Reviewer 3:** *"Guidelines[/Examples] for building such input and readout maps".*

Aside from the examples arising from classification and Riemannian Exponential/Logarithm map examples discussed in Sections 3.1 and 3.2, two examples of feature maps between Euclidean spaces, satisfying the Assumptions 3.1 and 3.2 are the following. Let $g : \mathbb{R}^m \to \mathbb{R}^d$ be a continuous function, then $\phi(x) \triangleq (x, g(x))$ satisfies Assumption 3.1. Alternatively, if $A_{i,j}$ are any full-rank square matrices if $j \neq 2$ and $A_{2,1} \circ A_{1,1}$ is well-defined, then the set of DNNs of the form $\rho \circ f \circ \phi$ where

$$\rho(x) = Leaky\text{-}ReLU \bullet (A_{1,K}x + b_{1,K}) \circ \cdots \circ Leaky\text{-}ReLU \bullet (A_{1,1}x + b_{1,1})$$
$$f(x) = [A_{2,2} \circ \bullet ReLU \bullet (A_{2,1}x + b_{2,1}) + b_{2,2}] \tag{1}$$
$$\phi(x) = Leaky\text{-}ReLU \bullet (A_{3,k}x + b_{3,k}) \circ \cdots \circ Leaky\text{-}ReLU \bullet (A_{3,1}x + b_{3,1}).$$

are universal since the input and output maps both satisfy Assumptions 3.1 and 3.2. Note that the matrices $A_{i,j}$ may be highly sparse with at-least $m$ (resp. $n$) non-zero entries for $\rho$ (resp. $\phi$).

As a class of non-examples, if $A_1, \ldots, A_K, B_1, \ldots, B_k$ are any square matrices and $C_2, C_1$ are composable matrices then, the set of DNNs of the form

$$ReLU \bullet (A_n x + b_n) \circ \cdots \circ ReLU \bullet (A_1 x + b_1) \tag{2}$$

are not universal and the input and output maps violate Assumptions 3.1 and 3.2.

**Deeper Layers -Reviewer 2:** *"[It's] common practice [to] change the internal structure of the architecture beyond the input and output layers...people often inject a particular inductive bias [into the DNN]".*

Examples (1) and (2) show that in a DNN, the matrices $A_{i,j}$ ($j \neq 2$) can be chosen as we like so-long as they are of full-rank. Therefore, for a DNN to be universal, we only need the middle two layers, described by $f$, to be fully-connected. In particular, this gives us the flexibility of encoding many "inductive biases" into the architecture since only the two middle layers cannot be modified freely, as long as the involved matices' ranks are preserved.

**Relation to Future Research -Reviewer 3:** *Example: Generalizability via Dropout but while maintaining approxima-tion capabilities.* Consider a DNN of the form $\rho \circ (A_{2,2} \circ \sigma \bullet (A_{1,2}x + b_{1,1}) + b_{2,2}) \circ \phi$ where $\phi$ and $\rho$ are as in 1, $B_i$ are arbitrary composable matrices and $c_i$ are vectors of appropriate dimension. Since 1 only requires defining $\rho$ and $\phi$ to be of full-rank but can be highly sparse. Therefore, Theorem 3.3 implies that if dropout is used to improve generalizability, it can only maintain universal approximation if the dropout procedure is constrained so that it preserves the matrix's rank. This is interesting, since the generalization effects of dropout are well-understood but the impact of drop-out on an architecture's approximation abilities, or more generally sparsely-connected DNNs, is so-far not.

**Numerical Illustrations/Experiments - Reviewers 1-4:** To illustrate the effect of properly (or poorly) choosing the networks' input and output layers we implement the architecture of (1) (Good), the architecture defined by (2) (Bad), and a shallow feed-forward network with no additional input and output maps (Vanilla) as a baseline model. Our implementations are on the California housing dataset [3], with the objective of predicting the median housing value, the test-set consists of $30\%$ of the total data, pre-processing as in [2] and would be included in the camera-ready version. As anticipated, applying a readout and feature map satisfying our conditions can only improves the performance of our the architecture by learning a good representation of the data. In contrast, a poorly chosen feature map degrades the model's performance.

| | Good | Bad | Vanilla | | Good | Bad | Vanilla |
|---|---|---|---|---|---|---|---|
| MAE: Test | 0.381672 | 2.073648 | 0.428122 | MAE: Train | 0.316039 | 5.637205 | 0.375244 |
| RMSE: Test | 0.420023 | 2.056685 | 0.434948 | RMSE: Train | 0.374318 | 5.548146 | 0.391001 |

**Illustration of Stakes - Reviewer 4:** *"One can hope that something as simple as the softmax function ... does not spoil the UA of the previous layers".* It is not surprising that the softmax function preserve's the ability for an architecture to approximate any classifier point-wise in $[0,1]^n$ and uniformly in $(0,1)^n$. However, point-wise convergence of a deep-classifier to any "non-fuzzy classifier" (taking values in $[0,1]^n - (0,1)^n$) is not robust since this means that selected network depends on the size of the training set, both in practice and in theory. Theorem 3.9 guarantees, amongst other things, that a single network can theoretically be trained which approximately performs the classification task with uniform precision on all of $X$ for any classifier, even the "non-fuzzy classifiers" taking values in $[0,1]^n - (0,1)^n$. Similar issues arise with the other mentioned examples and we would be happy to add a brief discussion outlining each.

[1] Guilbault, Craig R. and Tirel, Carrie J. On the dimension of Z-sets. Topology Appl., 160, 2013, 1.
[2] A. Geron, Handson-ML. Accessed: 2020-05-15.
[3] Kaggle. California housing prices. Accessed: 2020-05-15.


[Meta-Review · NeurIPS 2020]

Reviewers unanimously recognized the interest and value of this paper. In the final revision, please address all promises in the rebuttal and comments in the review. In particular, please try to expand the experiments---even simulated experiment may help understand the contribution; the rebuttal suggested some possibilities here that would help. Additionally, please add explanation in the background section about the manifold and topological terminology used.